# PASTA kinase-dependent control of peptidoglycan synthesis via ReoM is required for cell wall stress responses, cytosolic survival, and virulence in *Listeria monocytogenes*

Jessica L. Kelliher[1], Caroline M. Grunenwald[1], Rhiannon R. Abrahams[1], McKenzie E. Daanen[1], Cassandra I. Lew[2], Warren E. Rose[2], John-Demian Sauer[1]*

**1** Department of Medical Microbiology and Immunology, University of Wisconsin-Madison, Madison, Wisconsin, United States of America, **2** School of Pharmacy, University of Wisconsin-Madison, Madison, Wisconsin, United States of America

* sauer3@wisc.edu

## Abstract

Pathogenic bacteria rely on protein phosphorylation to adapt quickly to stress, including that imposed by the host during infection. Penicillin-binding protein and serine/threonine-associated (PASTA) kinases are signal transduction systems that sense cell wall integrity and modulate multiple facets of bacterial physiology in response to cell envelope stress. The PASTA kinase in the cytosolic pathogen *Listeria monocytogenes*, PrkA, is required for cell wall stress responses, cytosolic survival, and virulence, yet its substrates and downstream signaling pathways remain incompletely defined. We combined orthogonal phosphoproteomic and genetic analyses in the presence of a β-lactam antibiotic to define PrkA phosphotargets and pathways modulated by PrkA. These analyses synergistically highlighted ReoM, which was recently identified as a PrkA target that influences peptidoglycan (PG) synthesis, as an important phosphosubstrate during cell wall stress. We find that deletion of *reoM* restores cell wall stress sensitivities and cytosolic survival defects of a Δ*prkA* mutant to nearly wild-type levels. While a Δ*prkA* mutant is defective for PG synthesis during cell wall stress, a double Δ*reoM* Δ*prkA* mutant synthesizes PG at rates similar to wild type. In a mouse model of systemic listeriosis, deletion of *reoM* in a Δ*prkA* background almost fully restored virulence to wild-type levels. However, loss of *reoM* alone also resulted in attenuated virulence, suggesting ReoM is critical at some points during pathogenesis. Finally, we demonstrate that the PASTA kinase/ReoM cell wall stress response pathway is conserved in a related pathogen, methicillin-resistant *Staphylococcus aureus*. Taken together, our phosphoproteomic analysis provides a comprehensive overview of the PASTA kinase targets of an important model pathogen and suggests that a critical role of PrkA *in vivo* is modulating PG synthesis through regulation of ReoM to facilitate cytosolic survival and virulence.

**Data Availability Statement:** All relevant data are within the manuscript and its Supporting Information files.

**Funding:** This work was supported by National Institutes of Health grants awarded to J.L.K. (F32-AI154570), W.E.R. (R01-AI132627), and J.-D.S. (R01-AI137070 and R21-AI144060), a Burroughs Wellcome Fund Investigators in the Pathogenesis of Infectious Diseases Award (https://www.bwfund.org) awarded to J.-D.S., and an American Heart Association grant awarded to C.M.G. (AH830263). The funders had no role in study design, data collection and analysis, decision to publish, or preparation of the manuscript.

**Competing interests:** The authors have declared that no competing interests exist.

## Author summary

Many antibiotics target bacterial cell wall biosynthesis, justifying continued study of this process and the ways bacteria respond to cell wall insults during infection. Penicillin-binding protein and serine/threonine-associated (PASTA) kinases are master regulators of cell wall stress responses in bacteria and are conserved in several major pathogens, including *Listeria monocytogenes*, *Staphylococcus aureus*, and *Mycobacterium tuberculosis*. We previously showed that the PASTA kinase in *L. monocytogenes*, PrkA, is essential for the response to cell wall stress and for virulence. In this work, we combined proteomic and genetic approaches to identify PrkA substrates in *L. monocytogenes*. We show that regulation of one candidate from both screens, ReoM, increases synthesis of the cell wall component peptidoglycan and that this regulation is required for pathogenesis. We also demonstrate that the PASTA kinase-ReoM pathway regulates cell wall stress responses in another significant pathogen, methicillin-resistant *S. aureus*. Additionally, we uncover a PrkA-independent role for ReoM *in vivo* in *L. monocytogenes*, suggesting a need for nuanced modulation of peptidoglycan synthesis during infection. Cumulatively, this study provides new insight into how bacterial pathogens control cell wall synthesis during infection.

## Introduction

The mammalian cytosol is a restrictive environment to microbes [1]. Non-cytosol-adapted organisms, including vacuolar pathogens, that encounter the cytosol are killed [2–4], often through triggering innate immune pathways that have evolved to sense and dispense of mislocalized bacteria [5–14]. Therefore, professional cytosolic pathogens must possess adaptations to survive and replicate in the cytosol. However, relatively little is known about the host defense mechanisms that restrict and kill non-adapted bacteria in the cytosol or the adaptations that professional cytosolic pathogens possess to promote their survival. Thus, continued studies into the molecular factors that drive host-pathogen interactions in the cytosol are needed to improve our understanding of this infection interface.

*Listeria monocytogenes*, a well-studied model for cytosolic pathogenesis [15], is a Gram-positive saprophyte that can infect humans and livestock through contaminated food or water [16]. Young, old, pregnant, or immunocompromised individuals are especially susceptible to systemic listeriosis, a disease with a high mortality rate (20–30%) in cases that require hospitalization, despite antibiotic intervention [16,17]. Upon ingestion, *L. monocytogenes* invades through the intestinal epithelium, inducing its own uptake with internalin proteins. Subsequently, innate immune cells such as macrophages can phagocytose *L. monocytogenes* [18]. Once inside a cell, *L. monocytogenes* deploys its pore-forming cytolysin listeriolysin O (LLO, encoded by *hly*) to escape the vacuole and reach the cytosol [19]. Once in the cytosol, *L. monocytogenes* begins replicating and uses actin-based motility and two phospholipases to evade autophagy [20]. Actin propulsion also promotes spread to neighboring cells, where *L. monocytogenes* uses LLO and its phospholipases to puncture the new double-membrane vacuole and again reach the cytosol, repeating the cellular infection cycle [21]. To establish systemic disease, *L. monocytogenes* must access the cytosol, and maintain its intracellular niche, for replication and dissemination. While the molecular determinants of *L. monocytogenes* invasion, phagosomal escape, and actin-based motility have been well defined, the adaptations required for *L. monocytogenes* survival and replication in the cytosol remain sparsely characterized.

Previously, we found that the penicillin-binding protein and serine/threonine-associated (PASTA) kinase PrkA is essential for *L. monocytogenes* adaptation to the cytosol [22]. PASTA kinases are single-component signal transduction systems that sense muropeptide fragments through extracellular PASTA domains to monitor cell wall integrity [23–26] and transmit the activation signal to a eukaryotic-like serine/threonine kinase (eSTK) domain in the bacterial cytoplasm [27]. A *L. monocytogenes* mutant lacking *prkA* is more sensitive to a variety of cell envelope stressors *in vitro*, including β-lactam antibiotics and host-derived factors like lysozyme and LL-37 [22,28]. In addition to genetic inactivation, we have also found that pharmacological inhibition of PrkA renders *L. monocytogenes* more sensitive to β-lactam antibiotics and other cell wall-targeting insults *in vitro* [28,29]. In the context of infection, a ΔprkA mutant is defective in cytosolic survival and replication in macrophages *ex vivo* and is avirulent in a mouse model of listeriosis [22]. PASTA kinases are conserved in single copy in *Firmicutes* and Actinobacteria, and genetic or biochemical disruption of these kinases in other pathogens also results in increased sensitivity to β-lactam antibiotics and virulence attenuation, including in *Staphylococcus aureus* [30–33], *Enterococcus faecalis* [34], *Streptococcus pneumoniae* [35,36], *Bacillus anthracis* [37], and *Mycobacterium tuberculosis* [38]. Despite the critical, conserved role of PASTA kinases in a variety of pathogens and their attractive potential for antimicrobial targeting, the mechanisms through which these stress response systems regulate bacterial physiology remain poorly understood.

In contrast to traditional two-component phosphorelay systems in bacteria, PASTA kinases phosphorylate multiple targets [27,39]. Phosphoproteomic and other systematic studies in *B. subtilis* [40], *S. aureus* [41,42], *M. tuberculosis* [43,44], and others [45,46] have revealed several common PASTA kinase-regulated pathways, including central carbon metabolism, nucleotide metabolism, and virulence-associated processes such as biofilm formation. Most prominent among PASTA kinase-regulated processes, however, is cell wall synthesis and homeostasis, which is often targeted at different nodes within the same species. For example, Stk1 of *S. aureus* modulates cell wall composition via positive phosphoregulation of the two-component systems GraRS, which increases expression of the *dltABCD* operon and therefore D-alanylation of wall teichoic acids [47], and WalRK, which regulates several autolysins important for cell division [48]. In addition, metabolomic analysis in *S. aureus* strongly suggests that the kinase regulates peptidoglycan (PG) synthesis, as metabolite levels in this pathway are significantly decreased in a Δ*stk1* mutant [41]. In *M. tuberculosis*, the PASTA kinase PknB phosphorylates CwlM, which in turn stimulates activity of MurA, the enzyme that catalyzes the first step in PG synthesis [49]. PknB also directly phosphorylates GlmU, the enzyme that produces the substrate of MurA, UDP-GlcNAc [50]. Another common PASTA kinase substrate in Gram-positives involved in cell wall homeostasis is GlmR (formerly YvcK), a protein of unknown function that has been proposed in *B. subtilis* to influence muropeptide precursor synthesis through regulation of GlmS, which catalyzes conversion of fructose-6-phosphate to glucosamine-6-phosphate [51]. We previously found that GlmR (YvcK) is a PrkA substrate in *L. monocytogenes* and is essential for cell wall stress responses, cytosolic survival, and virulence [22].

The only other validated *L. monocytogenes* PrkA phosphosubstrate, recently identified by Wamp *et al.* [52], is ReoM (regulator of MurA degradation). This small (90 AA) protein is conserved among PASTA kinase-containing *Firmicutes*, including *Enterococcus faecalis*, where the first ReoM homolog, IreB, was identified as a kinase substrate involved in intrinsic β-lactam resistance [53]. While the precise function of ReoM homologs is yet unknown, in *L. monocytogenes*, phosphorylation of ReoM by PrkA prevents ClpCP-mediated degradation of MurA, the enzyme that catalyzes the first committed step in muropeptide synthesis [52]. Thus, PrkA regulation of ReoM results in increased flux through MurA and therefore increased PG synthesis. Deletion of *reoM*, *clpC*, *clpP*, or another small protein of unknown function in the pathway,

*reoY*, phenocopies phosphorylation of ReoM and similarly stabilizes MurA levels [52]. Introduction of a phosphoablative ReoM$_{T7A}$ variant, which cannot be phosphorylated by PrkA, reduces MurA protein levels and increases sensitivity of *L. monocytogenes* to the β-lactam ceftriaxone [52], suggesting that phosphoregulation of ReoM by PrkA is important for the response to cell wall stress. However, whether PrkA regulation of ReoM impacts the intracellular lifecycle and virulence of *L. monocytogenes* remains unexplored.

While previous studies have sought to define the phosphoproteome in *L. monocytogenes* [54,55] or to identify PrkA-interacting proteins [56], a systematic identification of PrkA phosphosubstrates in the context of cell wall stress and/or virulence is lacking. In this study, we sought to identify adaptations controlled by PrkA during cell wall stress, pairing phosphoproteomic analysis and a suppressor screen to identify targets and downstream pathways of PrkA in the context of β-lactam stress. These analyses synergistically revealed that ReoM is a critical PASTA kinase substrate in *L. monocytogenes* during β-lactam exposure. Targeted genetic analysis revealed that loss of *reoM* reverses sensitivity of a Δ*prkA* mutant to a variety of cell wall envelope stressors and restores PG synthesis defects. Deletion of *reoM* almost completely reversed *ex vivo* and *in vivo* virulence defects of a Δ*prkA* mutant, suggesting that ReoM is also a key substrate of PrkA in the context of infection. Indeed, all isolates sequenced from an *in vivo* suppressor screen mapped to the pathway controlling PG synthesis beginning with PrkA-mediated phosphorylation of ReoM. Importantly, however, loss of *reoM* alone reduced virulence *ex vivo* and *in vivo*, indicating that negative control of PG synthesis by ReoM is also critical at some point during the infectious lifecycle of *L. monocytogenes*. We found that PASTA kinase-mediated regulation of ReoM was also important for intrinsic resistance to β-lactams in the pathogen *S. aureus*. Cumulatively, these findings define the phosphosubstrates of PrkA in *L. monocytogenes* during cell wall stress and demonstrate that PrkA-mediated control of PG synthesis in the cytosol via phosphorylation of ReoM is critical for the virulence of this dangerous pathogen.

## Results

### *prkA* is not an essential gene in *L. monocytogenes*

In many pathogens, PASTA kinases contribute to the response to cell wall stress, intrinsic β-lactam resistance, and virulence. To identify targets and pathways modulated by PrkA in *L. monocytogenes*, we performed parallel phosphoproteomic and suppressor analyses in the context of cell wall stress. First, we set out to define the phosphoproteome of wild-type *L. monocytogenes* and a conditional Δ*prkA* mutant (Δ*prkA*$_{cond}$, described previously [28]) in which *prkA* is deleted from its native locus and a *trans*-encoded copy of *prkA* is controlled by a theophylline-inducible riboswitch. Wild type and Δ*prkA*$_{cond}$ were exposed to a sub-inhibitory concentration of the β-lactam ceftriaxone (CRO), to which the Δ*prkA*$_{cond}$ mutant is ~100-fold more sensitive than wild type in the absence of inducer [22]. This preliminary phosphoproteomic analysis revealed 241 unique arginine, aspartic acid, cystine, glutamic acid, histidine, lysine, serine, threonine, or tyrosine phosphopeptides in wild-type *L. monocytogenes* in the presence of CRO (S1 Table). Combining phosphopeptides that differed only by missed cleavages, oxidations, or deamidations, 23 unique phosphosites were absent or significantly less abundant in the Δ*prkA*$_{cond}$ mutant, suggesting they may be direct PrkA targets (S1 Table).

Of note, one of the top ten most abundant phosphopeptides found in the Δ*prkA*$_{cond}$ mutant samples mapped to PrkA, suggesting that the inducible riboswitch controlling the expression of *prkA* is leaky. Multiple independent groups have constructed conditional Δ*prkA* mutations due to an inability to make an unmarked deletion, suggesting, as is the case in *M. tuberculosis*, that the PASTA kinase PrkA is essential. To determine if PrkA is essential and to minimize phosphorylation events by PrkA in our phosphoproteomics experiments, we attempted to

**Table 1. MICs of cell envelope-targeting agents against Δ*reoM* strains in BHI.**

| Strain | Ampicillin | Vancomycin | Bacitracin | Lysozyme | LL-37 |
|---|---|---|---|---|---|
| WT | 0.125 (±0) | 1 (1–2) | 125 (125–250) | >2000 (±0) | 100 (100->200) |
| Δ*prkA*::*erm* | 0.0156 (0.0078–0.0156) | 0.5 (±0) | 62.5 (62.5–125) | 500 (±0) | 6.25 (6.25–25) |
| Δ*reoM* | 0.25 (0.125–0.25) | 1 (1–2) | 125 (125–250) | >2000 (±0) | >200 (±0) |
| Δ*reoM* Δ*prkA*::*erm* | 0.125 (±0) | 1 (1–2) | 125 (125–250) | >2000 (±0) | >200 (±0) |
| WT pPL2 | 0.125 (±0) | 1 (1–2) | 125 (125–250) | >2000 (±0) | 100 (100->200) |
| Δ*prkA*::*erm* pPL2 | 0.0156 (0.0078–0.0156) | 0.5 (±0) | 62.5 (62.5–125) | 500 (500–1000) | 6.25 (6.25–25) |
| Δ*reoM* pPL2 | 0.25 (0.125–0.25) | 1 (1–2) | 125 (±0) | >2000 (±0) | >200 (50->200) |
| Δ*reoM* pPL2-P$_{reoM}$-*reoM* | 0.125 (±0) | 1 (1–2) | 125 (±0) | >2000 (±0) | 100 (50->200) |
| Δ*reoM* Δ*prkA*::*erm* pPL2 | 0.125 (±0) | 1 (±0) | 125 (125–250) | >2000 (±0) | >200 (±0) |
| Δ*reoM* Δ*prkA*::*erm* pPL2-P$_{reoM}$-*reoM* | 0.0313 (0.0156–0.0313) | 1 (±0) | 125 (±0) | 1000 (1000->2000) | 6.25 (6.25->50) |

Values are the median MICs of three biological replicates (and range of values) reported in µg/ml.

construct a new strain with a marked deletion mutant of *prkA* replaced with an erythromycin resistance cassette under a constitutive promoter, Δ*prkA*::*erm*, without introducing a *trans*-encoded copy of *prkA*. Unlike in the case of an unmarked deletion, we were able to recover Δ*prkA*::*erm* mutants, suggesting that PrkA is not essential. In contrast to what we have previously observed for the Δ*prkA*$_{cond}$ strain, the Δ*prkA*::*erm* mutant had a slight growth defect in BHI compared to wild type (S1A Fig), perhaps suggesting why construction of unmarked clean deletions of *prkA* have been previously unsuccessful. This result also supports the notion that some amount of *prkA* is expressed in the Δ*prkA*$_{cond}$ strain in the absence of inducer. Consistent with a critical role for PrkA in cell envelope stress responses, the Δ*prkA*::*erm* mutant was similarly sensitive as Δ*prkA*$_{cond}$ to a variety of stressors, including CRO, ampicillin, lysozyme, and LL-37 (S1B Fig and Table 1). Expression of *prkA* from a constitutive promoter *in trans* restored both the growth defect in BHI (S1C Fig) and intrinsic resistance to CRO (S1D Fig) of the Δ*prkA*::*erm* mutant.

Our previous studies on *prkA* had required the use of a conditional mutant [22,28] and Wamp *et al*. [52] recently reported that *prkA* is essential in *L. monocytogenes*. Therefore, to ensure that our marked Δ*prkA*::*erm* mutant was not recovered due to second site suppressor mutations, we transduced the Δ*prkA*::*erm* allele into a fresh wild-type 10403S background and performed whole-genome sequencing on 10 randomly selected transductants (Δ*prkA*::*erm* T1-T10) in addition to the Δ*prkA*::*erm* parent. The Δ*prkA*::*erm* parent contained one mutation in *Lmrg_02080* (Table 2), annotated as *lieB*, a putative ABC transporter permease. One transductant contained a six base pair deletion at the *Lmrg_02823* locus, encoding a putatively PG-anchored protein (LPXTG-containing) of unknown function. In two transductants, the prophage encoded at the *comK* locus had excised from the genome (Table 2). Importantly, in the remaining seven transductants, no mutations were detected relative to the parent wild-type strain, demonstrating that our ability to make the marked Δ*prkA*::*erm* deletion strain was not due to compensatory secondary mutations elsewhere in the genome. All 10 transductants grew similarly to the Δ*prkA*::*erm* parent both in BHI (S1A Fig) and in a CRO MIC assay (S1B Fig). We conclude therefore that *prkA* is not an essential gene in *L. monocytogenes*.

## Phosphoproteomic analysis reveals PrkA-dependent phosphorylations during β-lactam stress

We selected Δ*prkA*::*erm* T1 to perform phosphoproteomic analysis because this transductant contained no mutations relative to wild type other than the deletion of *prkA*. Wild-type and

**Table 2. Mutations in Δ*prkA*::*erm* strain and 10 Δ*prkA*::*erm* transductants compared to parent wild-type 10403S.**

| Isolate | Locus | Name | Mutation | Variant | Function/Description |
|---|---|---|---|---|---|
| Δ*prkA*::*erm* | Lmrg_02080 | lieB | G993534A | G221R | Permease subunit of ABC transporter |
| Δ*prkA*::*erm* T-1 | | | - | | |
| Δ*prkA*::*erm* T-2 | Lmrg_02823 | | GGCATCC1827309G | ADA480A | Putative peptidoglycan-bound protein (LPXTG motif-containing) |
| Δ*prkA*::*erm* T-3 | Lmrg_01560 | comK | | Prophage excision | Natural competence regulatory gene |
| Δ*prkA*::*erm* T-4 | Lmrg_01560 | comK | | Prophage excision | Natural competence regulatory gene |
| Δ*prkA*::*erm* T-5 | | | - | | |
| Δ*prkA*::*erm* T-6 | | | - | | |
| Δ*prkA*::*erm* T-7 | | | - | | |
| Δ*prkA*::*erm* T-8 | | | - | | |
| Δ*prkA*::*erm* T-9 | | | - | | |
| Δ*prkA*::*erm* T-10 | | | - | | |

Δ*prkA*::*erm* T1 cultures were treated with half an MIC of CRO and harvested at mid-log phase. Lysates were subjected to phosphopeptide enrichment, followed by LC-MS/MS analysis for phosphoproteome determination. To fully characterize the phosphoproteome of *L. monocytogenes*, we searched for arginine, aspartic acid, cystine, glutamic acid, histidine, lysine, serine, threonine, or tyrosine phosphorylations, and identified 127 unique phosphopeptides in wild-type *L. monocytogenes* present during β-lactam stress (S2 Table). We next separately searched for only serine and threonine phosphorylation events in wild-type and Δ*prkA*::*erm* samples to identify likely PrkA targets (Fig 1A and S3 Table). We defined PrkA-dependent phosphopeptides as those that were significantly ($P$ value < 0.05) less abundant in the Δ*prkA* mutant compared to in wild type. Using this cutoff, we identified 39 PrkA-dependent phosphopeptides (Fig 1A and S3 Table). Combining phosphopeptides that differed only by missed cleavages, oxidations, or deamidations, there were 26 unique phosphorylation sites across 23 proteins (Fig 1B). Among this list of putative PrkA substrates were *Listeria* homologs of several confirmed or putative PASTA kinase substrates in other organisms, namely HPr [57], ReoM [42,52,53], GpsB [58], and HtrA [42] (Fig 1B). The remaining 18 proteins represented novel PASTA kinase substrates. The proteins identified cover a broad range of biological functions, among them cell division (GpsB, FtsH), metabolism (ROK family glucokinase, CysK), and stress response (HtrA, MnSOD, catalase) (Fig 1C, left panel). They include cytoplasmic, membrane-associated, and ribosomal proteins (Fig 1C, middle panel). At least twelve of the putative substrates have catalytic activity, and several have receptor and signal-transducing activity (Fig 1C, right panel). These data are thus consistent with a role for PrkA having pleiotropic effects on bacterial physiology during cell wall stress in *L. monocytogenes*.

## Mutations suppressing cell wall stress sensitivity of a Δ*prkA* mutant also restore virulence-associated defects

In parallel with the initial phosphoproteomics experiment, we performed a screen for mutations that could suppress the sensitivity of the Δ*prkA*$_{cond}$ mutant to CRO to identify pathways modulated by PrkA during cell wall stress. We constructed an EMS-mutagenized library of the Δ*prkA*$_{cond}$ mutant, targeting ~1 mutation/genome as calculated by a rifampicin resistance assay [59]. $10^6$ CFU of the library was plated on agar plates containing 2 μg/ml CRO, which completely inhibited the growth of the parent strain Δ*prkA*$_{cond}$ but not wild type. We isolated 21 CRO-resistant suppressors for further analysis, naming them Δ*prkA*$_{cond}$ CRO Suppressor (PCS) mutants. Because the Δ*prkA*$_{cond}$ parent strain contains a *trans*-encoded *prkA* allele

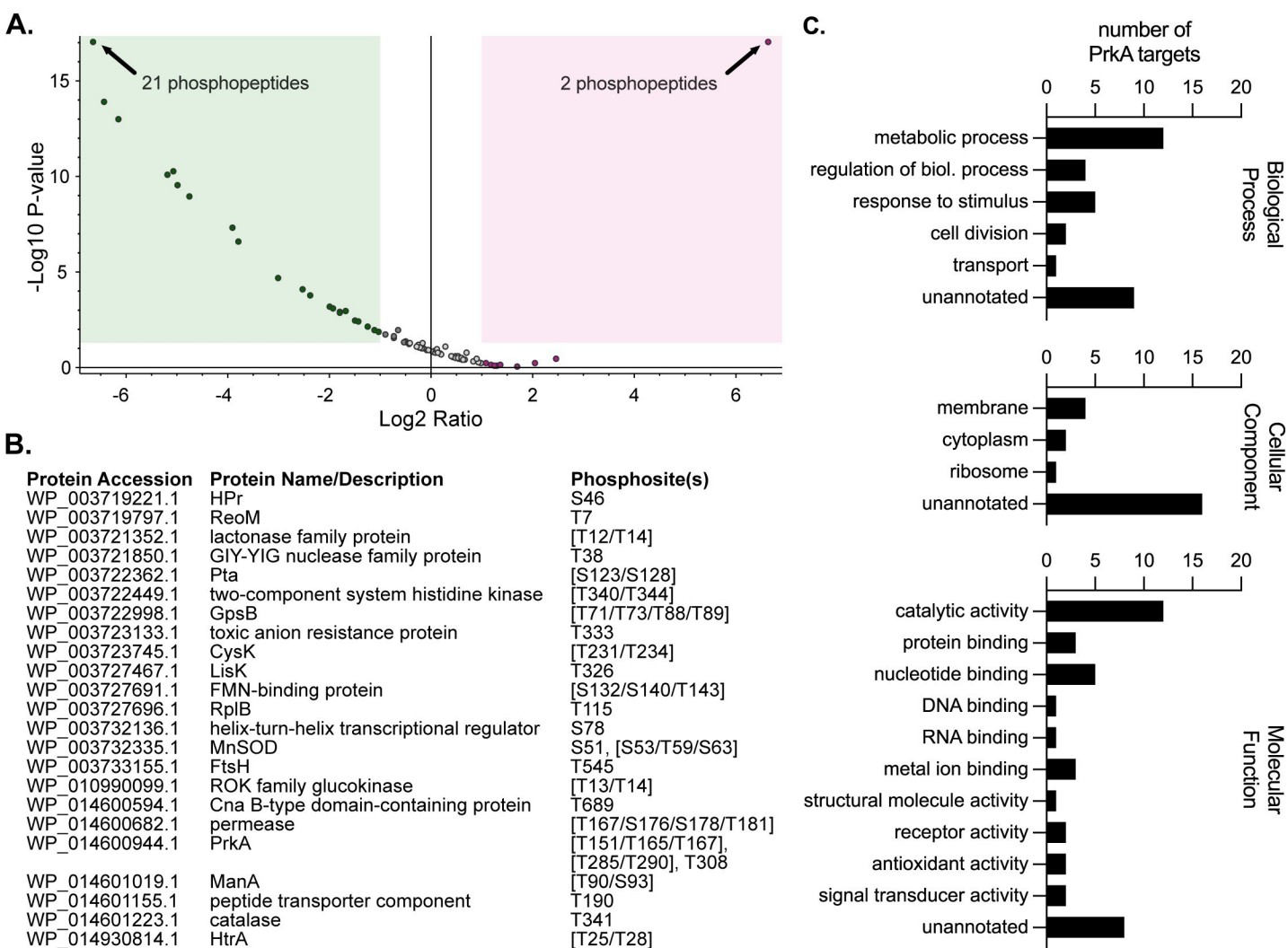

**Fig 1. PrkA phosphorylates proteins involved in diverse processes in the context of cell wall stress.** (**A**) Volcano plot showing log2 fold-change of abundance ratios versus -log10 *P* value of phosphorylated peptides with XCorr scores > 2.0 in the Δ*prkA*::*erm* mutant compared to wild type. The green box highlights peptides that were two-fold or less abundant and significantly different (*P* < 0.05) and the red box highlights peptides that were two-fold or more abundant and significantly different in the Δ*prkA*::*erm* mutant; some dots represent more than one phosphopeptide (indicated). (**B**) List of proteins containing phosphosites that were significantly less abundant in the Δ*prkA*::*erm* mutant compared to wild type. Differentially phosphorylated peptides from (**A**) that differed only by oxidations, deamidations, and/or missed cleavages were combined into the same phosphosite. Amino acids in brackets indicate ambiguous phosphosites. (**C**) GO term analysis of the proteins from (**B**). Some proteins are listed in more than one annotation per graph.

under the control of a theophylline-inducible riboswitch [28], we initially sequenced the riboswitch in the 21 PCS mutants to first rule out riboswitch mutations, which we presumed would result in increased *prkA* expression and be the cause of suppression. Of the 21 mutants, 9 (43%) contained riboswitch mutations. We moved forward with analysis of three randomly selected PCS mutants with wild-type riboswitch sequences: PCS2, PCS3, and PCS16. To confirm that these suppressors are indeed more resistant to CRO than the Δ*prkA*$_{cond}$ parent, we measured the MIC of CRO for PCS2, PCS3, and PCS16. The MIC of CRO was restored to that of, or within two-fold of, wild type for all three suppressors (Fig 2A). The three suppressor mutants also had MICs for ampicillin (AMP) comparable to wild type (Table 3). Two of the three PCS mutants were additionally more resistant to lysozyme than Δ*prkA*$_{cond}$ with the exception of PCS16, which was 8-fold more sensitive (Table 3).

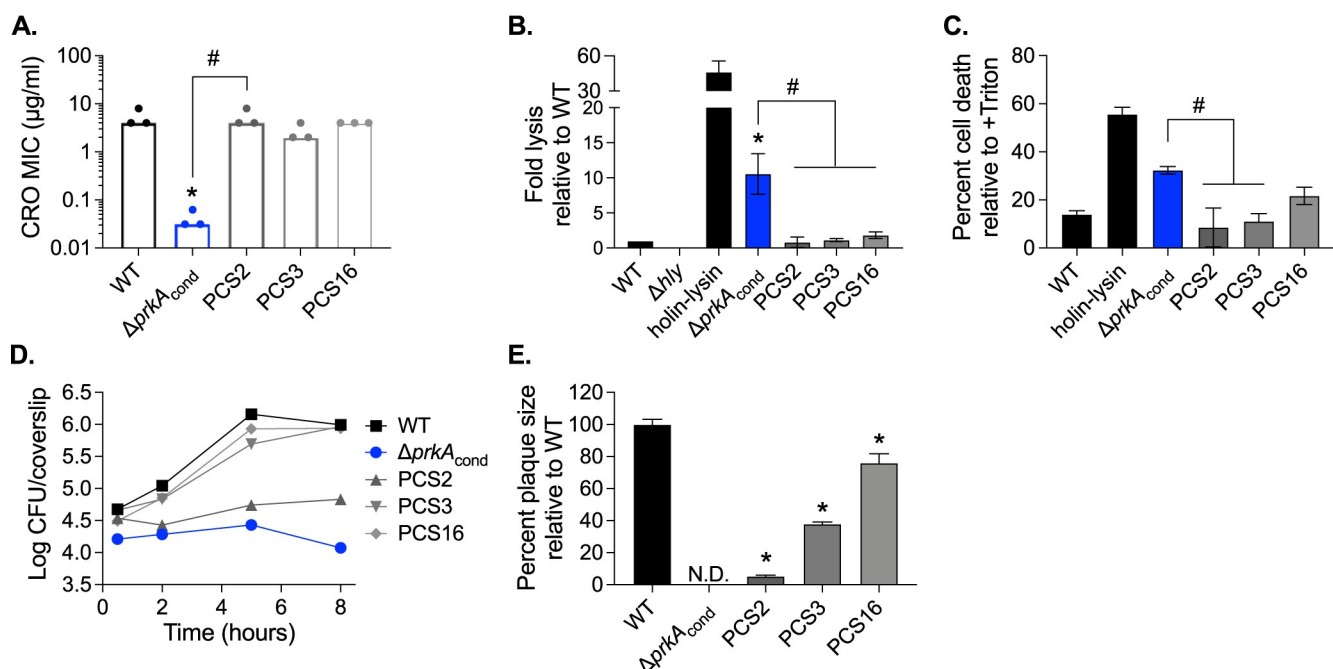

**Fig 2. Suppressors of CRO sensitivity also suppress *ex vivo* virulence defects of a Δ*prkA* mutant.** (A) Bars indicate median MICs of CRO for the indicated strains; n = 3. *, $P < 0.05$ compared to wild type, and #, $P < 0.05$ for the indicated comparisons, by one-way ANOVA with Tukey's multiple comparisons test. (B) Intracellular bacteriolysis in immortalized *Ifnar*$^{-/-}$ macrophages. Macrophages were infected with the indicated strains carrying the pBHE573 reporter vector at an MOI of 10, and luciferase activity was measured 6 hours post-infection. Error bars indicate standard error of the mean (SEM); n = 5. (C) Host cell death in primary C57BL/6 bone marrow-derived macrophages (BMDMs). BMDMs were infected with the indicated strains at an MOI of 10, and lactate dehydrogenase activity in cell supernatants was measured 6 hours post-infection. Error bars indicate SEM; n = 3. (B-C) *, $P < 0.05$ compared to wild type, and #, $P < 0.05$ compared to Δ*prkA*$_{cond}$ by one-way ANOVA with Tukey's multiple comparisons test. (D) Intracellular growth in C57BL/6 BMDMs. BMDMs seeded on glass coverslips were infected with the indicated strains at an MOI of 10, and CFU per coverslip was enumerated at the indicated time points. Data are representative of three biological replicates. (E) Plaque formation in immortalized murine fibroblasts (L2 cells). L2s were infected with the indicated strains at an MOI of ~0.5, and plaque sizes were normalized to those of wild type on day 6 of infection. Error bars indicate SEM; data are averaged from a minimum of 15 plaques from three biological replicates. N.D., not detected. *, $P < 0.05$ compared to wild type by one-way ANOVA with Tukey's multiple comparisons test.

We next assessed whether the three PCS mutants selected for their ability to suppress cell wall stress sensitivity could also relieve the severe phenotypes of a Δ*prkA* mutant during infection. Specifically, the Δ*prkA*$_{cond}$ parent strain lyses more frequently than wild type in the cytosol of host cells, which induces more inflammasome-mediated death of host cells [22]. To measure cytosolic killing of the PCS mutants compared to Δ*prkA*$_{cond}$, we employed a luciferase-based reporter system previously described [6]. In this assay, *L. monocytogenes* carries a plasmid-borne luciferase gene under the control of a host-driven promoter, so that luciferase is only produced by the infected cell upon release of the plasmid from lysed bacteria and

**Table 3. MICs of cell envelope-targeting agents against PCS mutants in BHI.**

| Strain | Ampicillin | Lysozyme |
|---|---|---|
| WT | 0.125 (0.125–0.25) | >4096 (±0) |
| Δ*prkA*$_{cond}$ | 0.0078 (±0) | 512 (±0) |
| PCS2 | 0.0625 (0.0625–0.125) | 2048 (±0) |
| PCS3 | 0.0625 (0.0625–0.125) | >4096 (±0) |
| PCS16 | 0.0625 (0.0625–0.125) | 64 (±0) |

Values are the median MICs of three biological replicates (and range of values) reported in μg/ml.

luciferase activity can be quantified to measure relative cytosolic bacteriolysis. A strain lacking LLO, the pore-forming cytolysin that enables escape from the vacuole, does not reach the cytosol and thus does not report in this assay (Fig 2B). A strain engineered to express two *Listeria* phage lysis proteins upon entry into the cytosol (holin-lysin) lyses at high levels, resulting in high luciferase activity (Fig 2B). Consistent with prior observations, the $\Delta prkA_{\text{cond}}$ mutant lysed at ~10-fold higher levels than wild-type *Lmo* (Fig 2B). All three PCS mutants displayed significantly less bacteriolysis than $\Delta prkA_{\text{cond}}$, restoring lysis levels to that near wild type (Fig 2B). Correspondingly, while the $\Delta prkA_{\text{cond}}$ mutant caused ~3-fold higher cell death than wild-type *L. monocytogenes* as measured by LDH release, the PCS mutants caused less cell death than the $\Delta prkA_{\text{cond}}$ mutant (Fig 2C).

Consistent with prior observations [22], the $\Delta prkA_{\text{cond}}$ mutant was also unable to replicate in the cytosol of macrophages; indeed, bacterial burden was lower at the end of the growth curve than the start, supporting the idea that this mutant is killed in the cytosol (Fig 2D). In contrast, PCS3 and PCS16 were able to replicate at rates similar to wild type and reach similar burdens to wild type after 8 hours (Fig 2D). Growth of PCS2 was restricted, with bacterial numbers remaining static over the course of the growth curve (Fig 2D). Notably, however, PCS2 burdens did not decrease like those of the $\Delta prkA_{\text{cond}}$ parent (Fig 2D). Consistent with previous results [22], the $\Delta prkA_{\text{cond}}$ mutant was unable to form plaques in a monolayer of fibroblasts while wild type does so robustly. Plaque formation requires phagosomal escape, bacterial replication in the cytosol, and cell-to-cell spread. All three PCS mutants were able to plaque, albeit to differing degrees (Fig 2E). PCS16 formed almost wild-type-sized plaques, while PCS3 and especially PCS2 had diminished plaquing ability, forming plaques ~40% and ~5% of the size of wild type, respectively (Fig 2E). The minimal size of plaque formation by the PCS2 mutant is likely attributable to its slow intracellular growth (Fig 2D). Cumulatively, these data demonstrate that mutations that suppress sensitivity to cell wall-targeting antibiotics can restore some virulence-related phenotypes of a $\Delta prkA$ mutant, suggesting that cell wall stress responses regulated by PrkA are critical for *L. monocytogenes* virulence.

## PCS mutations map to the ReoM pathway controlling MurA stability

To identify the mutations responsible for suppression, PCS2, PCS3, and PCS16 were submitted for whole genome sequencing. Six mutations relative to the $\Delta prkA_{\text{cond}}$ parent were found across the three genomes, two of which were common to all three mutants (Table 4). PCS2 had an additional unique missense mutation in the proteolytic subunit *clpP*, and PCS3 had a unique missense mutation in the ATPase subunit *clpC* (Table 4). Of note, *clpP* and *clpC* mutants alone have defects in intracellular replication [60,61], consistent with our observations that PCS2 and PCS3 only partially restore $\Delta prkA_{\text{cond}}$ defects in intracellular growth and plaque formation (Fig 2D and 2E). PCS16 additionally contained mutations in *Lmrg_02076*, a GNAT-family N-acetyltransferase, and 12 base pairs upstream of the *reoM* start codon (Table 4). Presumably, the mutation upstream of *reoM* reduced its expression, similar to a $\Delta gpsB$ suppressing mutation in the RBS of *reoM* recently isolated by Wamp *et al.* that is phenocopied by deletion of the *reoM* CDS [52]. Thus, each of the three PCS isolates contained a mutation in the ReoM pathway that controls MurA degradation, suggesting that PrkA-mediated phosphorylation of ReoM is important for the response to cell wall stress both *in vitro* and *ex vivo*.

We next sequenced the genomes of the remaining 9 PCS mutants with wild-type riboswitch sequences to identify other potential pathways modulated by PrkA in the presence of CRO. These 9 mutants contained 1 to 6 mutations relative to the $\Delta prkA_{\text{cond}}$ parent. Three PCS mutants contained mutations in *clpC*, two mutants contained mutations in *clpP*, four mutants contained mutations in the CDS or immediately upstream of *reoM*, and three mutants

**Table 4. Mutations in PCS mutants compared to the $\Delta prkA_{cond}$ parent strain.**

| Isolate | Locus | Name | Mutation | Variant | Function/Description |
|---------|-------|------|----------|---------|----------------------|
| PCS1 | Lmrg_01600 | | G2282263A | E337K | Hypothetical protein |
| | Lmrg_01695 | murZ | C2583617A | Intergenic (39 bp upstream of start) | MurA paralog |
| PCS2 | Lmrg_00741 | oatA | C1277896T | Silent | Peptidoglycan O-acetyltransferase |
| | Lmrg_01780 | clpP | G2496832A | R112H | Catalytic subunit of Clp protease |
| | Lmrg_01945 | | G2789642A | A241T | ATP-binding subunit of multidrug efflux pump |
| PCS3 | Lmrg_02674 | clpC | C246587A | Q512K | ATPase subunit of Clp protease |
| | Lmrg_00741 | oatA | C1277896T | Silent | Peptidoglycan O-acetyltransferase |
| | Lmrg_01945 | | G2789642A | A241T | ATP-binding subunit of multidrug efflux pump |
| PCS6 | Lmrg_00741 | oatA | C1277896T | Silent | Peptidoglycan O-acetyltransferase |
| | Lmrg_01695 | murZ | C2582699CT | Frameshift, premature stop codon | MurA paralog |
| | Lmrg_01945 | | G2789642A | A241T | ATP-binding subunit of multidrug efflux pump |
| | Lmrg_01919 | | C2822334A | Silent | Hypothetical protein |
| PCS7 | Lmrg_01780 | clpP | G2496732A | D79N | Catalytic subunit of Clp protease |
| PCS8 | Lmrg_00641 | cbiE | C1180142T | P96S | Precorrin-6B methylase |
| | Lmrg_00741 | oatA | C1277896T | Silent | Peptidoglycan O-acetyltransferase |
| | Lmrg_01467 | reoM | A1491250AT | Frameshift, premature stop codon | Regulator of MurA degradation |
| | Lmrg_05020 | | G1699725A | | tRNA-Asn |
| | Lmrg_01834 | sufD | C2440473T | D45N | iron-sulfur cluster assembly protein |
| | Lmrg_01945 | | G2789642A | A241T | ATP-binding subunit of multidrug efflux pump |
| PCS14 | Lmrg_02674 | clpC | C245359G | Y102* | ATPase subunit of Clp protease |
| | Lmrg_00741 | oatA | C1277896T | Silent | Peptidoglycan O-acetyltransferase |
| | Lmrg_01945 | | G2789642A | A241T | ATP-binding subunit of multidrug efflux pump |
| PCS16 | Lmrg_02076 | | C989513T | G133E | GNAT family N-acetyltransferase |
| | Lmrg_00741 | oatA | C1277896T | Silent | Peptidoglycan O-acetyltransferase |
| | Lmrg_01467 | reoM | C1491331T | Intergenic (12 bp upstream of start) | Regulator of MurA degradation |
| | Lmrg_01945 | | G2789642A | A241T | ATP-binding subunit of multidrug efflux pump |
| PCS17 | Lmrg_02608 | gpmA | G284397T | Intergenic (19 bp upstream of start) | Phosphoglycerate mutase |
| | Lmrg_01467 | reoM | A1491250AT | Frameshift, premature stop codon | Regulator of MurA degradation |
| | Lmrg_02829 | rplT | C1815571T | G23E | 50S ribosomal protein 20 |
| PCS19 | Lmrg_00339 | | C677634T | E70K | GNAT family N-acetyltransferase |
| | Lmrg_00367 | flhB | C695925T | Silent | Flagellar biosynthesis |
| | Lmrg_00741 | oatA | C1277896T | Silent | Peptidoglycan O-acetyltransferase |
| | Lmrg_01695 | murZ | C2582788T | G262E | MurZ paralog |
| | Lmrg_01945 | | G2789642A | A241T | ATP-binding subunit of multidrug efflux pump |
| PCS20 | Lmrg_01467 | reoM | T1491349C | Intergenic (30 bp upstream of start) | Regulator of MurA degradation |
| PCS21 | Lmrg_02674 | clpC | C246587A | Q512K | ATPase subunit of Clp protease |
| | Lmrg_00741 | oatA | C1277896T | Silent | Peptidoglycan O-acetyltransferase |
| | Lmrg_01761 | | G2516839A | Silent | DUF-containing protein |
| | Lmrg_01723 | mreB | G2557908GA | Frameshift, premature stop codon | Rod shape-determining protein |
| | Lmrg_01945 | | G2789642A | A241T | ATP-binding subunit of multidrug efflux pump |

Shading represents SNPs in genes involved in the ReoM pathway characterized by Wamp *et al.* [52].

contained mutations in the CDS or immediately upstream of *murZ*. MurZ is a MurA paralog, and through an unknown mechanism, loss of function *murZ* mutations also result in increased MurA stability [62]. While an intergenic SNP could affect a small RNA or other regulatory element, we postulate that the SNP in PCS1 upstream of *murZ* decreases *murZ* expression, similar to a SNP upstream of *reoM* that decreases *reoM* expression [52]. Therefore, each of the twelve

PCS mutants sequenced contained a mutation in *reoM*, *clpC*, *clpP*, or *murZ* (Table 4). Of note, while 8 of the PCS mutants contained mutations in both *Lmrg_00741* and *Lmrg_01945*, the remaining 4 mutants did not, suggesting that these mutations were not required to compensate for disruption of the ReoM pathway. Furthermore, two PCS mutants, PCS7 and PCS20, contained only one mutation relative to the $\Delta prkA_{cond}$ parent strain, suggesting that disruption of the ReoM pathway and concomitant stabilization of MurA is sufficient for rescuing growth of the $\Delta prkA_{cond}$ mutant in the presence of CRO. Cumulatively, consistent with Wamp *et al.*, these findings suggest that the ReoM pathway is a critical PrkA-controlled adaptation to cell wall stress. Furthermore, our *ex vivo* virulence assays point to an important role for PrkA-mediated regulation of ReoM in promoting adaptation of *L. monocytogenes* to the eukaryotic cytosol.

### PrkA-mediated regulation of ReoM is required for PG synthesis during cell wall stress

ReoM was one of the most abundant PrkA-dependent phosphoproteins we identified in *L. monocytogenes* during exposure to CRO, and all PCS mutants sequenced contained mutations in the ReoM pathway that modulates MurA levels. Therefore, we chose to move forward with investigating the role of ReoM as a PrkA substrate during cell wall stress and infection. We constructed an in-frame deletion of *reoM* ($\Delta reoM$) and subsequently transduced the $\Delta prkA$::*erm* allele into the $\Delta reoM$ mutant ($\Delta reoM$ $\Delta prkA$::*erm*). The $\Delta reoM$ mutant was as resistant as wild type to all stressors tested and was slightly more resistant than wild type to CRO (Fig 3A), AMP, and LL-37 (Table 1). Strikingly, the double mutant $\Delta reoM$ $\Delta prkA$::*erm* restored the MIC to within 2-fold of wild type for CRO (Fig 3A), AMP, lysozyme, and LL-37 (Table 1). Complementation of the $\Delta reoM$ $\Delta prkA$::*erm* strain with a *trans*-encoded copy of *reoM* controlled by its native promoter (pPL2-P$_{reoM}$-*reoM*) restored sensitivity of the strain to all compounds tested (S2A Fig and Table 1). These results are consistent with the model put forth by Wamp *et al.* [52] and suggest that deletion of ReoM bypasses the requirement for phosphorylation by PrkA, preventing ClpCP-mediated degradation of MurA and resulting in increased PG synthesis to protect against cell wall insults.

Our findings together with the model put forth by Wamp *et al.* [52] suggests that $\Delta prkA$ mutants are defective in cell wall synthesis and that deletion of *reoM* in a $\Delta prkA$ background should restore this defect by increasing flux into the PG synthesis pathway. We directly tested this hypothesis by measuring PG labeling in wild-type, $\Delta prkA$::*erm*, $\Delta reoM$, and $\Delta reoM$ $\Delta prkA$::*erm* strains during cell wall stress as previously described [63,64]. Alkyne-conjugated D-alanine-D-alanine dipeptide (alk-D-ala-D-ala) was added to exponentially growing bacteria for 1 hour in the absence or presence of 0.5 MIC of CRO. The alk-D-ala-D-ala probe is incorporated into nascent PG and subsequently can be detected by copper-catalyzed azide-alkyne cycloaddition to a modified fluorophore (Fig 3B). In the absence of CRO, wild-type *L. monocytogenes* and $\Delta prkA$::*erm* were similarly labeled, suggesting that these strains synthesized similar amounts of PG in the absence of stress, consistent with a minimal growth defect of $\Delta prkA$::*erm* mutants in these conditions (Fig 3C). The $\Delta reoM$ mutant was significantly more labeled than wild type even in the absence of CRO, consistent with increased flux through MurA in the absence of *reoM* (Fig 3C) [52]. Upon exposure to CRO, labeling of wild-type *L. monocytogenes* increased to a similar magnitude as that of the $\Delta reoM$ mutant in the presence of CRO (Fig 3C), which was similar to its unstressed levels. In contrast, labeling of the $\Delta prkA$::*erm* mutant did not increase upon exposure to CRO (Fig 3C). Both in the absence and presence of CRO, the $\Delta reoM$ $\Delta prkA$::*erm* mutant was labeled at levels similar to that of the $\Delta reoM$ mutant (Fig 3C). These results strongly support the notion that PrkA is required for PG synthesis during cell

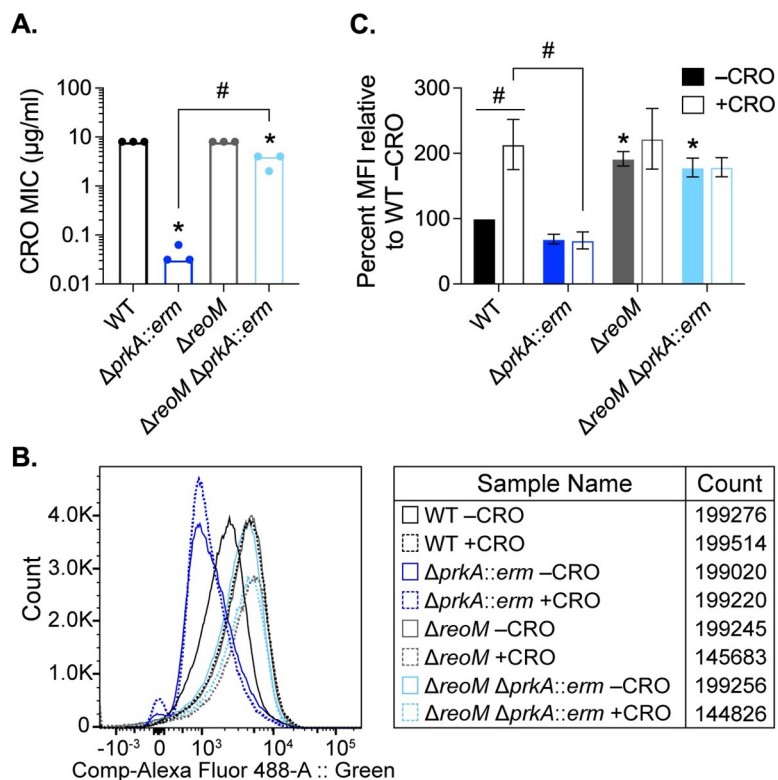

**Fig 3. Loss of *reoM* restores PG synthesis defects of a Δ*prkA* mutant of *L. monocytogenes* during cell wall stress.** (**A**) Bars indicate median MICs of CRO for the indicated *L. monocytogenes* strains; n = 3. *, $P < 0.05$ compared to wild type, and #, $P < 0.05$ for the indicated comparisons by one-way ANOVA with Tukey's multiple comparisons test. (**B-C**) The indicated strains of *L. monocytogenes* were grown in the presence of 1 mM alkyne-D-ala-D-ala for 1 hour in the absence and presence of 0.5 MIC of CRO. Cells were subjected to a copper-catalyzed alkyne-azide cycloaddition reaction with a modified Alexa Fluor 488, and fluorescence was subsequently analyzed by flow cytometry, with a minimum of $1.5 \times 10^5$ events read per sample. (**B**) Representative histograms for Alexa Fluor 488 signal. (**C**) Median fluorescence intensity (MFI) was normalized to that of WT minus CRO. *, $P < 0.05$ compared to wild type minus CRO by two-way ANOVA with Tukey's multiple comparison's test. #, $P < 0.05$ for the indicated comparisons by two-way ANOVA with Sidak's multiple comparison's test.

wall stress and that deletion of its phosphosubstrate ReoM restores PG synthesis defects of a Δ*prkA* mutant. In addition, these observations support a model in which PG synthesis is disrupted and constitutively high in an Δ*reoM* mutant.

## Deletion of *reoM* restores virulence-associated defects of a Δ*prkA* mutant

To assess whether PrkA-mediated regulation of ReoM is important for cytosolic survival, we measured intracellular bacteriolysis using the luciferase-based reporter system. Similar to the Δ*prkA*<sub>cond</sub> mutant [22], the Δ*prkA*::*erm* mutant lysed at significantly higher rates than wild-type *L. monocytogenes* (Fig 4A). The Δ*reoM* mutant alone lysed at similar rates to wild type, indicating inhibition of excessive peptidoglycan synthesis by ReoM is not necessary for cytosolic survival (Fig 4A). The Δ*reoM* Δ*prkA*::*erm* mutant also lysed at wild-type levels (Fig 4A), suggesting PrkA-mediated control of PG synthesis through ReoM is important for responding to cell wall stress in the cytosol. Re-introduction of *reoM* via pPL2-P<sub>reoM</sub>-*reoM* into the Δ*reoM* Δ*prkA*::*erm* mutant resulted in increased cytosolic killing (S2B Fig). As an *ex vivo* measure of virulence, we also assessed the ability of Δ*prkA*::*erm*, Δ*reoM*, and Δ*reoM* Δ*prkA*::*erm* to form plaques in a monolayer of fibroblasts. Like the Δ*prkA*<sub>cond</sub> mutant [22], the Δ*prkA*::*erm* mutant

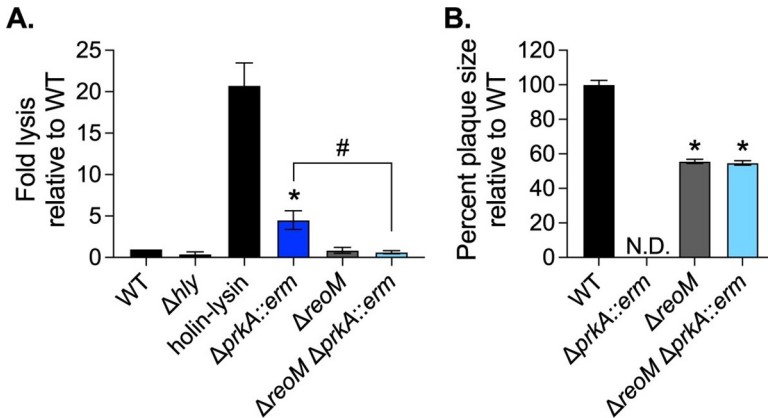

**Fig 4. Loss of *reoM* restores *ex vivo* virulence defects of a Δ*prkA* mutant.** (**A**) Intracellular bacteriolysis in immortalized *Ifnar*[-/-] macrophages. Macrophages were infected with the indicated strains carrying the pBHE573 reporter vector at an MOI of 10, and luciferase activity was measured 6 hours post-infection. Error bars indicate SEM; n = 4. *, $P < 0.05$ compared to wild type and #, $P < 0.05$ for the indicated comparison by one-way ANOVA with Tukey's multiple comparisons test. (**B**) Plaque formation in immortalized murine fibroblasts (L2 cells). L2s were infected with the indicated strains at an MOI of ~0.5, plaques were stained on day 4 of infection, and sizes were normalized to those of wild type. Error bars indicate SEM; data are averaged from a minimum of 81 plaques from three biological replicates. N.D., not detected. *, $P < 0.05$ compared to wild type by one-way ANOVA with Tukey's multiple comparisons test.

was unable to plaque (Fig 4B). The Δ*reoM* mutant alone exhibited a plaquing defect, forming plaques ~50% of the size of wild type (Fig 4B), and this defect was complemented by pPL2-P_{reoM}-*reoM* (S2C Fig). This finding implies that ReoM is required during the intracellular *L. monocytogenes* lifecycle and points to a potential role for ReoM in virulence. The Δ*reoM* Δ*prkA*::*erm* mutant partially suppressed the plaquing defect of Δ*prkA*::*erm*, forming plaques equivalent in size to the Δ*reoM* mutant (Fig 4B). Plaquing by the double mutant was substantially diminished by re-introduction of *reoM* via pPL2-P_{reoM}-*reoM* (S2C Fig). Of note, the complement strain Δ*reoM* Δ*prkA*::*erm* pPL2-P_{reoM}-*reoM* formed ~80% fewer plaques than the other strains, suggesting that reintroduction of *reoM* restored the Δ*prkA* mutant phenotype. The small number of plaques that did form in the Δ*reoM* Δ*prkA*::*erm* pPL2-P_{reoM}-*reoM* complement strain formed plaques roughly half the size of the Δ*reoM* Δ*prkA*::*erm* parent strain (S2C Fig). To account for this observation in our statistical analysis, we imputed values of 0 for plaque sizes up to the next fewest number of plaques formed by any strain in that experiment, which was Δ*reoM* pPL2-P_{reoM}-*reoM*. Altogether, these observations indicate that deletion of *reoM* rescues virulence phenotypes associated with loss of *prkA* in the cytosol, suggesting that PrkA-mediated control of PG synthesis is crucial in this environment. Additionally, these results suggest that ReoM is required for virulence independent of its role as a PrkA substrate.

## PrkA-mediated regulation of ReoM is critical for *L. monocytogenes* infection

We next investigated the impact of PrkA-mediated ReoM regulation on virulence of *L. monocytogenes*. Wild-type, Δ*prkA*::*erm*, Δ*reoM*, Δ*reoM* Δ*prkA*::*erm*, and the corresponding *reoM* complementation strains were used to infect C57BL/6 mice intravenously, and bacterial burdens were measured 48 hours post-infection. Consistent with our prior findings with the Δ*prkA*_{cond} mutant [22], CFU isolated from mice infected with the Δ*prkA*::*erm* mutant numbered at or below the limit of detection in both the spleen and the liver (Fig 5). The Δ*reoM* mutant was nearly as virulent as wild type in the spleen (Fig 5A) but exhibited a ~3-log defect

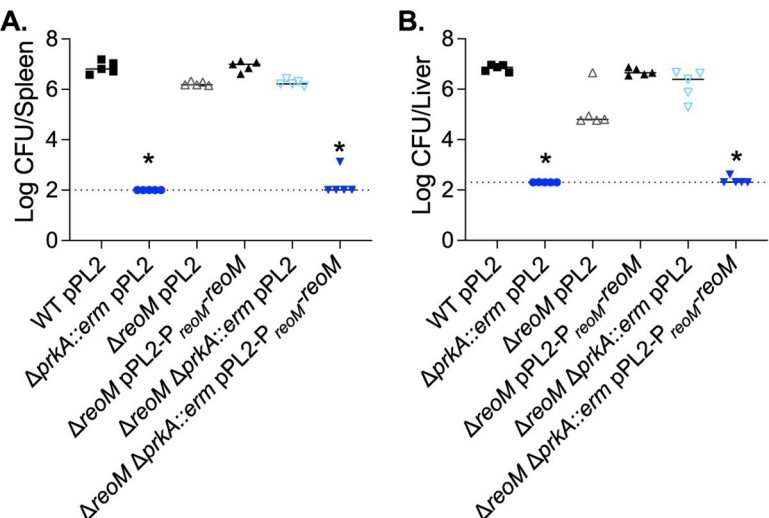

**Fig 5. PrkA-mediated regulation of ReoM is essential during infection.** (**A-B**) Nine-week-old C57BL/6 mice were infected intravenously with 1 x 10⁵ CFU of the indicated strains. Forty-eight hours post-infection, spleens (**A**) and livers (**B**) were harvested and CFU enumerated. Solid bars indicate medians and dotted lines indicate limits of detection. Data are representative of two independent experiments of 5 mice each. *, $P < 0.05$ compared to wild type by Kruskal-Wallis test with Dunn's multiple comparisons test.

in the liver, which could be complemented by pPL2-P$_{reoM}$-*reoM* (Fig 5B). Mice infected with the Δ*reoM* Δ*prkA*::*erm* mutant harbored nearly as many CFU as those infected with wild-type *L. monocytogenes* in both the spleen and the liver (Fig 5). The Δ*reoM* Δ*prkA*::*erm* mutant was attenuated by reintroduction of *reoM* via pPL2-P$_{reoM}$-*reoM* (Fig 5). These results suggest that increasing PG synthesis through deletion of *reoM* is sufficient to largely restore the severe virulence phenotype of a Δ*prkA* mutant, but that excessive PG synthesis caused by lack of ReoM in a wild-type strain is detrimental to virulence.

PASTA kinases are important for the response to myriad stressors and conditions that could be encountered in the host during infection [39]. Given that deletion of *reoM* did not fully restore virulence of the Δ*prkA* mutant *in vivo*, we hypothesized that other PrkA phosphorylation-dependent pathways may also contribute to pathogenesis. Therefore, we initiated an *in vivo* suppressor screen to gain insight into the adaptations that PrkA regulates during infection. Twelve wild-type mice were each infected with 1x10⁷ CFU of an EMS-mutagenized library of the Δ*prkA*::*erm* strain, representing ~100-fold coverage of each possible G:C to A:T transition in the *L. monocytogenes* genome in total. Seventy-two hours post-infection, spleens and livers were plated to enumerate CFU and isolate suppressors. Each organ contained CFU above the limit of detection, with some burdens approaching wild-type levels of ~10⁷ (S3 Fig). Following purification, one isolate from the spleen of each mouse, named Δ*prkA*::*erm* <u>v</u>irulence <u>s</u>uppressor (PEVS) mutants 1–12, was submitted for whole-genome sequencing. Two PEVS isolates had frameshift mutations in *reoM*, three isolates had mutations in *reoY*, and six isolates had frameshift mutations in *murZ*, all eleven of which resulted in premature stop codons (Table 5). The twelfth isolate contained a missense mutation in *murA* that did not map to the active site or other known regions of protein-protein interactions (Table 5). Thus, all twelve isolates contained a mutation in the ReoM pathway of MurA regulation, most of which were the only mutation relative to the Δ*prkA*::*erm* parent (8/12 isolates). Remarkably, although G:C to A:T transitions are present in the genomes of many of the *in vivo* suppressor isolates, none of the 12 mutations mapping to *reoM*, *reoY*, *murZ*, or *murA* are EMS-induced mutations, indicating that there is strong selective pressure in the context of the host to disrupt this

**Table 5. Mutations in PEVS mutants compared to the Δ*prkA*::*erm* parent strain.**

| Isolate | Locus | Gene | Mutation | Variant | Function/Description |
|---------|-------|------|----------|---------|---------------------|
| PEVS1 | *Lmrg_01695* | *murZ* | AT2583501A | Frameshift, premature stop codon | UDP-GlcNAc enolpyruvyl transferase (MurA paralog) |
| PEVS2 | *Lmrg_01467* | *reoM* | A1491250AT | Frameshift, premature stop codon | Regulator of MurA degradation |
| PEVS3 | *Lmrg_00558* | *guaA* | T1111632A | Intergenic (116 bp upstream of start) | GMP synthetase |
| | *Lmrg_01695* | *murZ* | CT2582699C | Frameshift, premature stop codon | UDP-GlcNAc enolpyruvyl transferase (MurA paralog) |
| PEVS4 | *Lmrg_01695* | *murZ* | CT2582699C | Frameshift, premature stop codon | UDP-GlcNAc enolpyruvyl transferase (MurA paralog) |
| PEVS5 | *Lmrg_01068* | *reoY* | T1953040A | K74* | Unknown function |
| PEVS6 | *Lmrg_01068* | *reoY* | T1953040A | K74* | Unknown function |
| PEVS7 | *Lmrg_01722* | *murA* | G2558560C | Q361E | UDP-GlcNAc enolpyruvyl transferase |
| | *Lmrg_01853* | | C2892783T | Silent | Putative MFS family transporter |
| PEVS8 | *Lmrg_01695* | *murZ* | T2582793TA | Frameshift, premature stop codon | UDP-GlcNAc enolpyruvyl transferase (MurA paralog) |
| PEVS9 | *Lmrg_02525* | *virA* | C1771939T | G605E | Permease subunit of ABC transporter |
| | *Lmrg_01068* | *reoY* | AT1952894A | Frameshift, premature stop codon | Unknown function |
| PEVS10 | *Lmrg_01695* | *murZ* | CT2582699C | Frameshift, premature stop codon | UDP-GlcNAc enolpyruvyl transferase (MurA paralog) |
| PEVS11 | *Lmrg_02602* | | G291232A | S48F | Hypothetical protein |
| | *Lmrg_00884* | | G1422502A | S192F | Hypothetical protein |
| | *Lmrg_01695* | *murZ* | CT2582699C | Frameshift, premature stop codon | UDP-GlcNAc enolpyruvyl transferase (MurA paralog) |
| PEVS12 | *Lmrg_01467* | *reoM* | A1491250AT | Frameshift, premature stop codon | Regulator of MurA degradation |

Shading represents SNPs in genes involved in the ReoM pathway characterized by Wamp *et al.* [52].

pathway in the absence of *prkA*. Cumulatively, these results suggest that regulation of ReoM by PrkA, which results in stabilization of MurA and increased PG synthesis, is a primary role of PrkA during *L. monocytogenes* infection.

## The PASTA kinase/ReoM pathway is conserved in *Staphylococcus aureus*

Originally identified in *E. faecalis*, ReoM homologs are conserved among Gram-positive PASTA-kinase containing organisms [65], and it was previously shown that the ReoM pathway is also intact in the model organism *Bacillus subtilis* [52]. To determine whether the function of ReoM and the downstream pathway that controls degradation of MurA is conserved and PASTA-kinase regulated in the important human pathogen *S. aureus*, we created an in-frame deletion of its *reoM* homolog (*SAUSA300_1574*). Consistent with previous findings [31,32], a mutant of *S. aureus* lacking the PASTA kinase Stk1 is more sensitive than wild type to the clinically relevant β-lactam antibiotic oxacillin (Fig 6A). The Δ*reoM* mutant alone was as

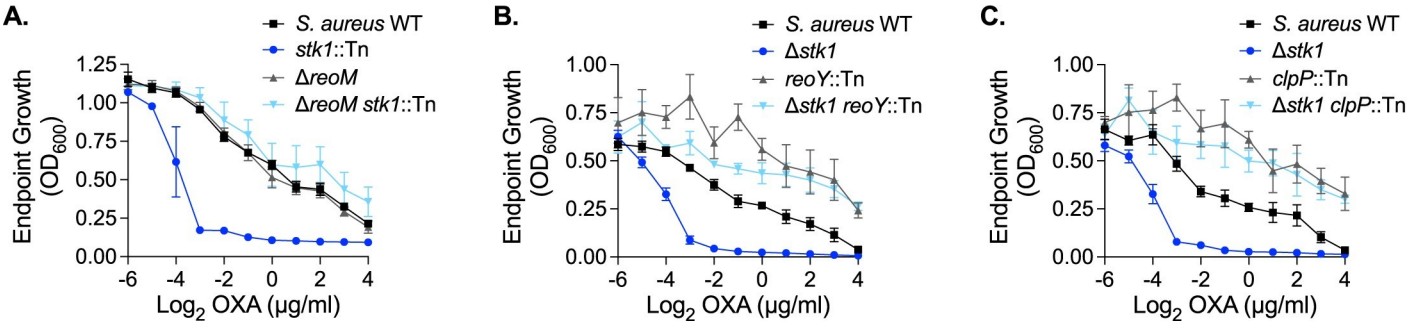

**Fig 6. The PASTA kinase/ReoM pathway is conserved in *S. aureus*.** The indicated *S. aureus* strains were grown in increasing concentrations of oxacillin for 12 hours. Endpoint growth measured by $OD_{600}$ is reported. Error bars indicate SEM; (**A**) n = 3, (**B**-**C**) n = 4.

resistant as wild type to oxacillin (Fig 6A). Similar to our findings in *L. monocytogenes*, deletion of *reoM* in the Δ*stk1* background restored resistance of the Δ*stk1* mutant to the cell wall-targeting antibiotic to wild-type levels (Fig 6A). We also determined that loss of *reoY* (*SAUSA300_1353*) (Fig 6B) or *clpP* (Fig 6C) restored intrinsic oxacillin resistance to the Δ*stk1* mutant, suggesting that the ReoM pathway functions similarly in *S. aureus* as it does in *L. monocytogenes*. Altogether, these findings suggest that ReoM is an important, conserved target of PASTA kinases upon activation by cell wall stress.

## Discussion

Pathogens must possess adaptations that promote their survival and growth in their infectious niche to avoid host defenses that target non-adapted invaders. We previously demonstrated that in *L. monocytogenes* a cell wall stress-sensing kinase, PrkA, is essential for coordinating adaptation to the cytosol and ultimately for virulence. Here, we set out to identify the phospho-substrates of PrkA important during cell wall stress and during infection, and report that PrkA-mediated regulation of the target ReoM is crucial *in vitro* and *in vivo* (Fig 7).

The necessity of the cell wall stress-sensing signal transduction system PrkA and downstream modulation of PG synthesis by phosphoregulation of ReoM for cytosolic survival implies that the host deploys cell wall-targeting insults in this niche. One major outstanding question is the identity of the host defenses that target cytosolic bacteria. While we have previously ruled out a role for GBPs [22], one host defense that could potentially explain the requirement of PrkA is lysozyme, which can target *L. monocytogenes* in the cytosol [66]. PCS16, which contained a SNP 12 base pairs upstream of the *reoM* start codon that likely decreases expression of *reoM*, was similarly resistant as wild type to CRO and AMP but was 8-fold more sensitive than the Δ*prkA*$_{cond}$ parent to lysozyme (Table 3). The cause of this may be another unique SNP in the PCS16 genome, a G133E mutation in *Lmrg_02076*, which encodes a putative GNAT-family acetyltransferase (Table 4). Notably, however, PCS16 displayed similar levels of intracellular bacteriolysis and replication as wild type (Fig 2B and 2D), suggesting that lysozyme was not responsible for cytosolic killing in these assays. Further studies are underway to identify cytosolic host factors that activate PrkA and necessitate modulation of PG synthesis.

This work offers several additional insights into the newly described ReoM-mediated MurA degradation pathway. Our data reveal that disruption of ReoM or other genes in the pathway can rescue the sensitivity of a Δ*prkA* mutant to a variety of cell wall-targeting antibiotics (e.g. CRO and AMP) and host-derived molecules (e.g. lysozyme and LL-37). PCS2 contained a R112H mutation in ClpP, which maps close to the active site of the enzyme based on its crystal structure [67]. PEVS7 contained a single missense mutation in *murA*. While this mutation did not map to the MurA active site or other known regions of protein-protein interactions, PEVS7 has a growth defect in rich media, suggesting this variation somehow affects function of the essential enzyme. We posit that this residue may also affect binding by ReoM, ReoY, and/or ClpCP and thus increase MurA stability. More detailed analysis of additional suppressors isolated from our *in vitro* and *in vivo* screens may provide mechanistic insight into ReoM and/or ReoY and how they interact with MurA to promote its degradation by ClpCP.

The strong selective pressure *in vivo* for disruption of the ReoM pathway in a Δ*prkA* background suggests that a majority of ReoM is usually phosphorylated by PrkA in the host. Why, then, maintain the ReoM pathway to control MurA stability? Our plaque assay and infection data revealed that loss of *reoM* on its own results in attenuation, with Δ*reoM* forming plaques half the size of wild type (Fig 4B) and exhibiting a 3-log defect in the liver (Fig 5B). These

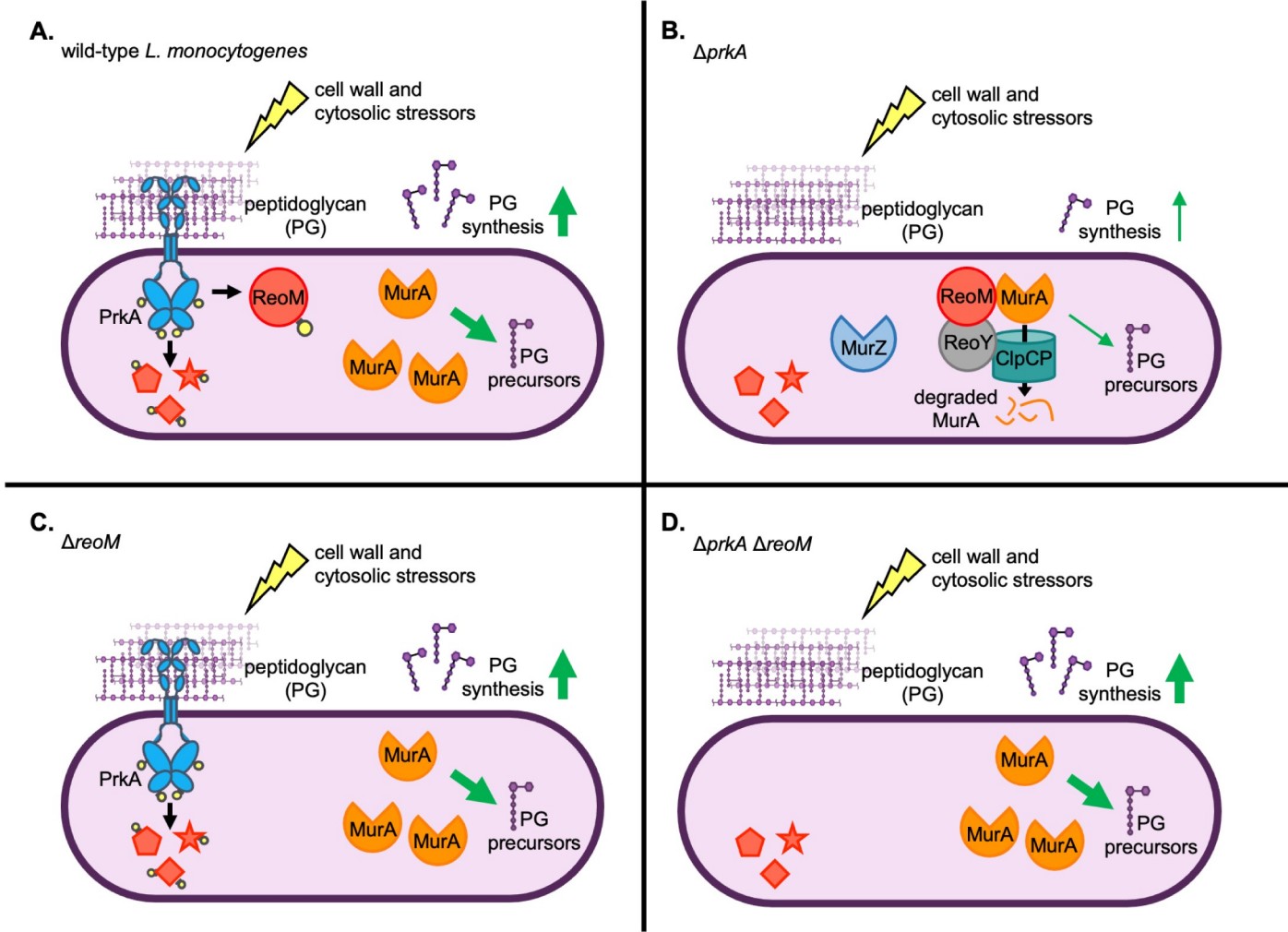

**Fig 7. Model: PrkA-mediated regulation of ReoM is required for peptidoglycan synthesis control during cell wall stress and in the cytosol.** (**A**) In wild-type *L. monocytogenes*, the PASTA kinase PrkA monitors cell wall integrity by sensing peptidoglycan (PG) perturbations. PrkA autophosphorylates to become activated (yellow circles indicate phosphorylations) by cell wall-targeting and cytosolic stressors and in turn phosphorylates substrates (represented by red shapes with yellow circles) that promote adaptation to cell wall stress and to the cytosol. Specifically, PrkA phosphorylates ReoM, which stabilizes the essential enzyme MurA by preventing targeting of MurA to the ClpCP protease. Increased MurA levels increases production of PG precursors, leading to increased PG synthesis. (**B**) In a Δ*prkA* mutant, PrkA substrates (red shapes) remain unphosphorylated in the presence of cell wall-targeting or cytosolic stressors. Unphosphorylated ReoM promotes ClpCP-dependent degradation of MurA and prevents an increase in PG synthesis. ReoY and MurZ also contribute to controlling MurA levels. Molecular interactions shown between ReoM, ReoY, MurA, and ClpCP are based on the findings of Wamp *et al.* [52]. (**C**) In a Δ*reoM* mutant, the absence of ReoM stabilizes MurA levels and increases PG synthesis (even in the absence of stress, not shown). PrkA can also phosphoregulate its other downstream targets in the presence of cell wall-targeting or cytosolic stressors. (**D**) In a Δ*prkA* Δ*reoM* mutant, lack of ReoM stabilizes MurA and leads to increased PG synthesis (even in the absence of stress, not shown), but other PrkA substrates cannot be phosphorylated during cell wall or cytosolic stress.

observations suggest that unphosphorylated ReoM is required for some stages of the intracellular lifecycle of *L. monocytogenes* and/or for some environments in the host. Interestingly, deletion of *prkA* in the Δ*reoM* background rescues the Δ*reoM* mutant *in vivo* (Fig 5B), suggesting that PrkA is toxic in the absence of ReoM. Our assessment of PG synthesis revealed that the Δ*reoM* mutant produces more PG than wild type, even in the absence of stress (Fig 3C). Of note, it is possible that our labeling approach could also reflect other changes in PG metabolism, such as changes in turnover of existing PG. However, our conclusion that Δ*reoM* mutants constitutively produce more PG is consistent with the previous observation that Δ*reoM* and Δ*reoY* mutants of *L. monocytogenes* have basally thicker cell walls [52]. While we did not

observe any growth defects with the Δ*reoM* mutant in rich medium or in the presence of cell wall stress, increased PG synthesis may lead to metabolic disruptions, i.e. aberrant misuse of intermediate metabolites, that are exacerbated by cell wall stress or during infection *in vivo*. Alternatively, increased cell wall thickness or other changes in PG composition may affect protein secretion, analogous to how PG branching affects pneumolysin release in *S. pneumoniae* [68]. An attempt to complement the Δ*reoM* mutant with *reoM* under the control of a constitutive, highly expressed promoter was unsuccessful, as all clones developed mutations that disrupted the complementary *reoM* CDS. This finding implies that native regulation of *reoM* expression is also important for routine growth. Cumulatively, these observations indicate that proper expression levels and phosphorylation status of ReoM are critical in standard laboratory growth conditions and *in vivo*. Further experimentation will be required to investigate the spatiotemporal factors that dictate the need for ReoM to dampen PG synthesis.

A significant difference between the findings of Wamp *et al.* [52] and those reported here is that we do not find PrkA or PrkA-mediated phosphorylation of ReoM to be essential for viability of *L. monocytogenes*. This discrepancy could potentially be due to strain differences between the 10403S and EGDe backgrounds. More likely, however, is that the fitness cost associated with loss of *prkA*, supported by the slight growth defect observed with the Δ*prkA*::*erm* strain (S1A Fig), makes generation of an unmarked Δ*prkA* using currently available tools improbable. The Δ*prkA*$_{cond}$ mutant we generated previously lacks *prkA* at the native locus but contains a *trans*-encoded, theophylline-inducible copy [28]. Leaving theophylline out of the growth medium does not result in any appreciable growth or morphological defects; however, our preliminary phosphoproteomics analysis using this mutant detected phosphorylated PrkA peptides, suggesting that this construct is minimally leaky (S1 Table). The ability to select for a marked deletion likely helped us overcome selective pressure that would arise against a *prkA* mutant during allelic exchange, and transduction of the allele does not involve an outgrowth step, limiting the ability of wild type to outcompete the mutant. Whole genome sequencing revealed that neither the marked deletion parent strain (Δ*prkA*::*erm*) nor 10 transductants contained mutations in the ReoM/PG synthesis pathway (Table 2). Indeed, most (7/10) of Δ*prkA*::*erm* sequenced transductants contained no mutations relative to wild type other than deletion of *prkA* (Table 2). We therefore conclude that the PASTA kinase in *L. monocytogenes* is not an essential gene. In this way, *L. monocytogenes* is similar to *B. subtilis*, *E. faecalis*, and *S. aureus*, in which the ReoM pathway is intact but the PASTA kinase is not essential [52,53,69,70] (Fig 6).

Consistent with PASTA kinase-dependent phosphoproteomes in other organisms [40,42,43,71], our analysis in *L. monocytogenes* supports the notion that PrkA broadly modulates bacterial physiology in response to cell wall stress. However, consistent with the existing literature, the overlap in putative PASTA kinase target lists among these organisms is somewhat less congruous, with only HPr, ReoM, GpsB, and HtrA homologs having been identified as PASTA kinase substrates in other bacteria previously [42,52,53,57,58]. Because phosphoproteins are much sparser in bacteria than in eukaryotes, bacterial phosphoproteomics experiments tend to be more variable across experiments and even across biological replicates. We find that many of the most differentially phosphorylated peptides in both our Δ*prkA*$_{cond}$ and Δ*prkA*::*erm* experiments are similar, suggesting our list of putative PrkA targets is robust. However, this list is likely not exhaustive, as it does not contain a previously identified PrkA substrate, YvcK [22]. Whether YvcK was not identified because it is a low abundance phosphopeptide or whether the conditions in which samples were prepared were not optimal for its phosphoregulation is unknown. Independent verification of PrkA phosphorylation of these putative targets is also needed, as not all phosphosites are likely to be direct substrates of PrkA. For example, S46 of HPr is phosphorylated by a dedicated HPr kinase [72], suggesting that

PrkA may have an indirect effect on phosphorylation of this site. Interestingly, in a phospho-proteomic analysis in *S. aureus*, phosphorylation S46 of HPr was found to be independent of Stk1, but a second serine residue, S294, was Stk1-dependent [42]. In *B. subtilis*, HPr is additionally phosphorylated at S12 by the PASTA kinase [57]. A second PrkA-dependent HPr phosphopeptide was not found in our datasets, suggesting PrkA may influence the phosphorylation state of HPr through an intermediate protein target. PrkA may have an indirect effect on a variety of other aspects of physiology, as two histidine kinases and a transcriptional regulator are among the putative PrkA targets (Fig 1B). While two-component system histidine kinases have been named putative PASTA kinase targets previously [42], the impact of serine/threonine phosphorylation of these sensor proteins remains unknown.

When taken together with the molecular studies of Wamp *et al.* [52], our findings strongly support a model in which ReoM is a critical PrkA substrate during cell wall stress and during infection (Fig 7). While our data support this model, we have not demonstrated PrkA-dependent phosphorylation of ReoM *in vivo* during infection, as current technical limitations of bacterial phosphoproteomics methods make such experiments challenging. Further studies are also needed to understand the importance of other PrkA targets in these contexts. Of note, while the Δ*reoM* mutant is strongly attenuated in the liver, loss of *prkA* in that background restores bacterial burdens nearly to the same level as wild type (Fig 5B). This finding implies that, at least in the absence of ReoM, PrkA phosphorylates another target(s) *in vivo* that negatively affects virulence. Similar to the apparent case with ReoM, it is possible that spatial and/or temporal cues dictate the need for phosphorylation of these other PrkA targets. Another possibility is that there is redundancy among PrkA targets such that phosphorylation of one of several substrates is sufficient to respond to cell wall stress, and constitutively high PG levels incurred upon loss of *reoM* masks the relevance of other targets. As in *B. subtilis* and others, PrkA targets at least two proteins involved in cell wall homeostasis: ReoM and GpsB, which controls function of the major PG synthesis enzyme penicillin-binding protein A1. Whether as in *B. subtilis* GpsB is a direct target of PrkA and the impact of PrkA-dependent regulation of GpsB on infection phenotypes is under investigation [58].

In conclusion, the findings herein provide a comprehensive view of the phosphosubstrates of PrkA during cell wall stress and demonstrate that phosphoregulation of ReoM is crucial for adaptation to the host cell cytosol. Future investigations into the ReoM pathway in infection is needed to understand its role in adapting the listerial cell wall to the intracellular lifestyle. Additionally, validation of other putative phosphosubstrates of PrkA identified in this study will inform additional ways that this important signal transduction system promotes the adaptation of *L. monocytogenes* to the cytosol. These studies have significant potential for identifying new avenues to develop antimicrobial therapies against important human pathogens.

## Materials and methods

### Ethics statement

All experiments involving animals were approved by the Institutional Animal Care and Use Committee of the University of Wisconsin-Madison.

### Bacterial strains, cloning, and growth conditions

*L. monocytogenes* was routinely grown at 37˚C in brain heart infusion (BHI) broth or on BHI + 1.5% agar plates and frozen in BHI + 40% glycerol. *S. aureus* was routinely grown at 37˚C in tryptic soy broth (TSB) or on TSB + 1.5% agar plates and frozen in BHI + 40% glycerol. *E. coli* was routinely grown at 37˚C in LB or on LB + 1.5% agar plates and frozen in LB + 40% glycerol. *E. coli* XL1-Blue was used for sub-cloning and SM10 or S17 were used for conjugating

**Table 6. Primers used in this study.**

| Name | Sequence | Description |
|------|----------|-------------|
| JLK57 | GCAGGTCGACTCTAGAGGATCCGTACCATTGACAAGGAAGAAAATGAAACG | F' for ~1 kb fragment upstream of *prkA* ORF |
| JLK55 | CCCGGCATCCGCTTACAGACAGAAGCATCCCTCCCTTTCTGCGTCAGATC | R' for ~1 kb fragment upstream of *prkA* ORF |
| JLK56 | CAGAAAGGGAGGGATGCTTCTGTCTGTAAGCGGATGCCGGG | F' for *erm* cassette with promoter |
| JLK4 | ACATTTCCTCCGTTCTATTTTTACTTATTAAATAATTTATAGCTATTGAAAAGAG | R' for *erm* cassette with promoter |
| JLK5 | ATAAATTATTTAATAAGTAAAAATAGAACGGAGGAAATGTGCTGGAAGG | F' for ~1 kb fragment downstream of *prkA* ORF |
| JLK58 | CGAGCTCGGTACCCGGGGATCCACGTCAATATGGATGTAATCTGCACCG | R' for ~1 kb fragment downstream of *prkA* ORF |
| JLK162 | GAAACCCATGGAAAAGGATCCATGATGATTGGTAAGCGATTAAGCGATC | F' for amplifying *prkA* ORF |
| JLK163 | CCGGGCCCCCCCTCGAGGTCGACTTAATTTGGATAAGGGACTGTACCTTCATC | R' for amplifying *prkA* ORF |
| JLK105 | CCCCTCGAGGTCGACGCGAAGTGAAACGTGAGAACG | F' for *reoM* and its native promoter |
| JLK106 | AGAACTAGTGGATCCTCATTTCTCACCAATTTCGTTATTTTTC | R' for *reoM* and its native promoter |
| CMG55 | GAGGCCCTTTCGTCTTCAAGAATTCGTGCAACACATTTATTACATGC | F' for ~1 kb fragment upstream of *reoM* ORF |
| CMG56 | GTAACATAAATTGCGACACTCCTTTAATTAC | R' for ~1 kb fragment upstream of *reoM* ORF |
| CMG57 | AGTGTCGCAATTTATGTTACAACATAAAATTTTAGGAC | F' for ~1 kb fragment downstream of *reoM* ORF |
| CMG58 | TCTAGAGGATCCCCGGGTACCATGACAATAAAATAAATCAGCCTTC | R' for ~1 kb fragment downstream of *reoM* ORF |

plasmids into *L. monocytogenes*. Antibiotics were used at the following concentrations: 200 μg/ml streptomycin, 50 μg/ml gentamicin, 10 μg/ml chloramphenicol, 30 μg/ml kanamycin, 2 μg/ml erythromycin, 1 μg/ml anhydrotetracycline.

Primers used for this study are listed in Table 6, plasmids in Table 7, and strains in Table 8. In-frame deletions were generated in *L. monocytogenes* as previously described [73] using pKSV7-oriT [74]. The pKSV7-Δ*prkA*::*erm* plasmid was constructed by Gibson Assembly with linearized pKSV7 and 3 PCR-generated fragments: a ~1 kb fragment upstream of the *prkA* ORF, amplified with JLK57 and JLK55 from *L. monocytogenes* 10403S gDNA; an *erm* cassette with a constitutive promoter, amplified with JLK56 and JLK4 from pPL2e-*riboE*-*prkA* [28]; and a ~1 kb fragment downstream of the *prkA* ORF, amplified with JLK5 and JLK58 from 10403S gDNA. The pKSV7-Δ*reoM* plasmid was constructed by Gibson Assembly with linearized pKSV7 and a gBlock (IDT) comprised of ~500 bp up- and downstream of the *reoM* ORF. The pIMK2-*prkA* plasmid was constructed by Gibson Assembly with linearized pIMK2 [75]

**Table 7. Plasmids used in this study.**

| Plasmid | Description | Reference |
|---------|-------------|-----------|
| pKSV7 | *L. monocytogenes* allelic exchange vector containing oriT | [74] |
| pKSV7-Δ*prkA*::*erm* | For replacing *prkA* with erythromycin resistance cassette controlled by a constitutive promoter | This study |
| pIMK2-*prkA* | For expression of *prkA* under a constitutive promoter | This study |
| pKSV7-Δ*reoM* | For making in-frame deletion of *reoM* | This study |
| pJB38 | *S. aureus* allelic exchange vector | [77] |
| pJB38-Δ*reoM* | For making in-frame deletion of *reoM* | This study |
| pJB38-Δ*stk1* | For making in-frame deletion of *stk1* | [29] |
| pPL2 | Integrative empty vector | [76] |
| pPL2-P$_{reoM}$-*reoM* | For expression of *reoM* under its native promoter (P$_{reoM}$) | This study |
| pBHE573 | Luciferase reporter vector for bacteriolysis assays | [6] |
| pBHE573k | Luciferase reporter vector for bacteriolysis assays with a kanamycin resistance cassette | This study |

**Table 8. Strains used in this study.**

| *L. monocytogenes* strains | Genotype/Description | Reference |
|---|---|---|
| Wild type | *L. monocytogenes* strain 10403S | [93] |
| Holin-lysin | Phage lysis proteins holin and lysin controlled by P$_{actA}$ | [6] |
| Δ*prkA*$_{cond}$ | Δ*prkA* with theophylline-inducible pPL2e-*riboE-prkA* | [28] |
| PCS2 | EMS-mutagenized Δ*prkA*$_{cond}$ (see Table 4) | This study |
| PCS3 | EMS-mutagenized Δ*prkA*$_{cond}$ (see Table 4) | This study |
| PCS16 | EMS-mutagenized Δ*prkA*$_{cond}$ (see Table 4) | This study |
| WT pBHE573 | Wild type carrying pBHE573 | [6] |
| Δ*hly* pBHE573 | In-frame deletion of *hly* carrying pBHE573 | [6] |
| Holin-lysin pBHE573 | Holin-lysin carrying pBHE573 | [6] |
| Δ*prkA*$_{cond}$ pBHE573 | Δ*prkA*$_{cond}$ carrying pBHE573 | [22] |
| PCS2 pBHE573 | PCS2 carrying pBHE573 | This study |
| PCS3 pBHE573 | PCS3 carrying pBHE573 | This study |
| PCS16 pBHE573 | PCS16 carrying pBHE573 | This study |
| Δ*prkA*::*erm* | *prkA* CDS replaced with *erm* cassette via pKSV7-Δ*prkA*::*erm* | This study |
| Δ*prkA*::*erm* pIMK2-*prkA* | Δ*prkA*::*erm* with integrated pIMK2-*prkA* | This study |
| Δ*reoM* | In-frame deletion of *reoM* via pKSV7-Δ*reoM* | This study |
| Δ*reoM* Δ*prkA*::*erm* | Δ*prkA*::*erm* allele transduced into Δ*reoM* | This study |
| Δ*prkA*::*erm* pBHE573 | Δ*prkA*::*erm* carrying pBHE573 | This study |
| Δ*reoM* pBHE573 | Δ*reoM* carrying pBHE573 | This study |
| Δ*reoM* Δ*prkA*::*erm* pBHE573 | Δ*reoM* Δ*prkA*::*erm* carrying pBHE573 | This study |
| WT pPL2 | Wild type with integrated pPL2 | This study |
| Δ*prkA*::*erm* pPL2 | Δ*prkA*::*erm* with integrated pPL2 | This study |
| Δ*reoM* pPL2 | Δ*reoM* with integrated pPL2 | This study |
| Δ*reoM* pPL2-P$_{reoM}$-*reoM* | Δ*reoM* with integrated pPL2-P$_{reoM}$-*reoM* | This study |
| Δ*reoM* Δ*prkA*::*erm* pPL2 | Δ*reoM* Δ*prkA*::*erm* with integrated pPL2 | This study |
| Δ*reoM* Δ*prkA*::*erm* pPL2-P$_{reoM}$-*reoM* | Δ*reoM* Δ*prkA*::*erm* with integrated pPL2-P$_{reoM}$-*reoM* | This study |
| WT pPL2 pBHE573k | WT pPL2 carrying pBHE573k | This study |
| Δ*hly* pBHE573k | Δ*hly* carrying pBHE573k | This study |
| Holin-lysin pBHE573k | Holin-lysin carrying pBHE573k | This study |
| Δ*prkA*::*erm* pPL2 pBHE573k | Δ*prkA*::*erm* pPL2 carrying pBHE573k | This study |
| Δ*reoM* pPL2 pBHE573k | Δ*reoM* pPL2 carrying pBHE573k | This study |
| Δ*reoM* pPL2-P$_{reoM}$-*reoM* pBHE573k | Δ*reoM* pPL2-P$_{reoM}$-*reoM* carrying pBHE573k | This study |
| Δ*reoM* Δ*prkA*::*erm* pPL2 pBHE573k | Δ*reoM* Δ*prkA*::*erm* pPL2 carrying pBHE573k | This study |
| Δ*reoM* Δ*prkA*::*erm* pPL2-P$_{reoM}$-*reoM* pBHE573k | Δ*reoM* Δ*prkA*::*erm* pPL2-P$_{reoM}$-*reoM* carrying pBHE573k | This study |
| ***S. aureus* strains** | **Genotype/Description** | **Reference** |
| Wild type | *S. aureus* USA300 JE2 | [79] |
| *stk1*::Tn | *stk1*::Tn allele from NTML library (NE217) transduced into WT JE2 | [79], this study |
| Δ*reoM* | In-frame deletion of *reoM* via pJB38-Δ*reoM* in JE2 background | This study |
| Δ*reoM stk1*::Tn | *stk1*::Tn allele transduced into Δ*reoM* | This study |
| Δ*stk1* | In-frame deletion of *stk1* via pJB38-Δ*stk1* in JE2 background | This study |
| *reoY*::Tn | *reoY*::Tn allele from NTML library (NE840) transduced into WT JE2 | [79] |
| Δ*stk1 reoY*::Tn | *reoY*::Tn allele transduced into Δ*stk1* | This study |

(*Continued*)

**Table 8.** (Continued)

| clpP::Tn | clpP::Tn allele from NTML library (NE912) transduced into WT JE2 | [79] |
|---|---|---|
| Δstk1 clpP::Tn | clpP::Tn transduced into Δstk1 | This study |

and a PCR fragment of the *prkA* CDS amplified from 10403S gDNA using primers JLK162 and JLK163. The pPL2-P*reoM*-*reoM* plasmid was constructed by Gibson Assembly with linearized pPL2 [76] and a PCR fragment of *reoM* and its native promoter amplified from 10403S gDNA using primers JLK105 and JLK106. The pJB38-Δ*reoM* plasmid was constructed using Gibson Assembly with pJB38 [77] linearized with restriction enzymes EcoRI and KpnI and PCR fragments of ~1kb up and downstream of the *reoM* ORF amplified with primer pairs CMG55/CMG56 and CMG57/CMG58, respectively, from genomic DNA purified from *S. aureus* USA300 JE2. All plasmid inserts were verified by Sanger sequencing. Phage transduction in *L. monocytogenes* was performed as previously described [78] using U153 phage. *S. aureus* transposon mutants were generated by transducing transposons from the Nebraska Transposon Mutant Library [79] into our laboratory stock of wild-type USA300 JE2. Phage transduction in *S. aureus* was performed as previously described [79,80] using φ85 phage.

## Sample preparation for phosphoproteome analysis

*L. monocytogenes* strains for phosphoproteome determination were grown overnight (16–18 hours) in BHI at 37˚C with shaking. Cultures were back-diluted 1:50 into 50 ml BHI containing half an MIC of ceftriaxone (see Table 4) in a 250 ml beveled flask and grown at 37˚C with shaking to an $OD_{600}$ of ~0.5. Cultures were pelleted at 3220 x *g* for 10 minutes at 4˚C, washed once in an equal volume of PBS, and pelleted again. Pellets were resuspended in 1 ml buffer (50 mM Tris, pH 8.0, 10 mM DTT, 0.1% SDS, 100 μM PMSF, phos-stop tablet [Roche]), and cells were lysed by 2 minutes of vortexing with zirconia beads (BioSpec Products). Lysates were pelleted by spinning at 21,130 x *g* for 10 minutes at 4˚C, and the supernatant was filtered through a 0.22 μm PES filter and stored at -80˚C.

## Enzymatic "in liquid" digestion and phosphoenrichment

Clarified and filtered lysates from *L. monocytogenes* were TCA/acetone precipitated (10% TCA, 40% acetone final) for protein extraction then pellets ([330 μg) re-solubilized and denatured in 100 μl of 8 M urea/50 mM $NH_4HCO_3$ (pH 8.5)/1 mM Tris-HCl. 2.5 μl of 1 mg/ml BSA was spiked as an internal standard and samples were diluted to 300 μl final volume with 160 μl of 25 mM $NH_4HCO_3$ (pH 8.5), 25 μl of Methanol, and 12.5 μl of 25 mM DTT. Samples were sonicated in a sonication bath for 3 minutes then incubated at 56˚C for 15 minutes during the reduction step, subsequently cooled on ice to room temperature, then 15 μl of 55 mM IAA was added for alkylation and incubated in darkness at room temperature for 15 minutes. The reaction was quenched by adding 40 μl of 25 mM DTT and subsequently 5 μl of 100 mM $CaCl_2$ and 25 μl of Trypsin stock was added (400 ng/μl *Trypsin Platinum* from Promega Corp. in 100 mM Tris-HCl [pH 8]) plus 115 μl of 25 mM $NH_4HCO_3$ (pH 8.5) to 500 μl final volume. Digestion was conducted overnight at 37˚C. Reaction was terminated by acidification with 2.5% TFA (Trifluoroacetic Acid) to 0.3% final concentration and samples were diluted to 800 μl total volume with 24 μl of 10% HFBA and 206 μl of milliQ water (final concentrations of 0.3% HFBA, 0.22% TFA and 3% methanol). These samples were desalted and concentrated using Strata-X 33 μm Polymeric Reverse Phase SPE cartridges (Phenomenex) per manufacturer protocol, where in short, methanol conditioned and water equilibrated cartridges had

samples applied in 2x 400 µl aliquots, washed quickly with milliQ water and eluted in 2x 300 µl of 70/29/1% ACN/$H_2O$/FA then dried to ~20 µl in the speed-vac and brought back to 150 µl total volume with 1 M glycolic acid in 80% ACN and 5% TFA, loading buffer. Phosphoenrichments were carried out in two stages, first using 1 mg of titanium dioxide magnetic microparticles (MagReSyn $TiO_2$ from ReSyn Biosciences) then unbound material was enriched with 1 mg of Zirconium-ion ($Zr^{4+}$) functional magnetic microparticles (MagReSyn Zr-IMAC from ReSyn Biosciences) and combined with the initial enriched eluate. Sample handling for both enrichments was according to the manufacturer protocols and evaluation of enrichments was internally standardized with a 5 pmol spike per each digest of AngioII-phosphate peptide standard.

### NanoLC-MS/MS and data analysis

Samples were analyzed on Orbitrap Fusion Lumos Tribrid platform, where 3 µl of each enriched sample was injected using Dionex UltiMate3000 RSLCnano delivery system (ThermoFisher Scientific) equipped with an EASY-Spray electrospray source (held at constant 50˚C). Chromatography of peptides prior to mass spectral analysis was accomplished using capillary emitter column (PepMap C18, 2 µM, 100 Å, 500 x 0.075 mm, Thermo Fisher Scientific). A NanoHPLC system delivered solvents A: 0.1% (v/v) formic acid, and B: 80% (v/v) acetonitrile, 0.1% (v/v) formic acid at 0.30 µL/min to load the peptides at 2% (v/v) B, followed by quick 1 minute gradient to 5% (v/v) B and gradual analytical gradient from 5% (v/v) B to 37.5% (v/v) B over 76 minutes followed by sharper gradient from 37.5% (v/v) B to 95% (v/v) B over 25 minutes when it concluded with a 5-minute flash-out at 95% (v/v) B. As peptides eluted from the HPLC-column/electrospray source survey MS scans were acquired in the Orbitrap with a resolution of 120,000 followed by HCD-type MS2 fragmentation into Ion Trap (32% collision energy) under ddMSnScan 1 second cycle time mode with peptides detected in the MS1 scan from 350 to 1600 m/z; redundancy was limited by dynamic exclusion and MIPS filter mode ON.

Lumos acquired MS/MS data files were searched using Proteome Discoverer (2.4.1.15) Sequest HT search engine against *Listeria monocytogenes* 10403S database (2,797 total sequences) plus cRAP common lab contaminant database (116 sequences). Static cysteine carbamidomethylation, variable methionine oxidation plus asparagine, glutamine deamidation, and serine, threonine phosphorylation with 2 tryptic miss-cleavages and peptide mass tolerances set at 10 ppm with fragment mass at 0.6 Da were selected. Peptide and protein identifications were accepted under strict 1% FDR cut offs with high confidence XCorr thresholds of 1.9 for z = 2 and 2.3 for z = 3. Strict principles of parsimony were applied for protein grouping. Chromatograms were aligned for feature mapping and area-based quantification using unique and razor peptides. Samples were normalized to spiked-in AngioII-phosphate. Phosphopeptides were filtered for "contaminant", "Quan Values", and "Phospho" modifications. Gene ontology (GO) terms were acquired for proteins in Proteome Discoverer.

### CRO suppressor isolation

The Δ$prkA_{cond}$ strain was mutagenized by a 10-minute exposure to 17.8 mM ethyl methanesulfonate (EMS) following a protocol previously described [59,81]. $10^6$ CFU of this library was plated on BHI agar plates containing 2 µg/ml CRO and grown at 37˚C. Twenty-one colonies were picked either 24 or 48 hours post-plating and screened secondarily on 2 µg/ml CRO plates (all 21 isolates grew on the second selection plate). Isolates were subsequently picked into 96-well plates containing 100 µl BHI broth, grown overnight at 37˚C, pelleted by centrifugation, resuspended in 100 µl BHI + 40% glycerol, and stored at -80˚C.

## Whole genome sequencing and SNP identification

*L. monocytogenes* isolates for whole-genome sequencing were grown overnight (16–18 hours) in BHI. Genomic DNA was purified using the MasterPure Gram-positive DNA purification kit (Epicentre) per the manufacturer's instructions, except that 5 U/μl mutanolysin was used instead of lysozyme. DNA was submitted to the Microbial Genome Sequencing Center (MiGS, Pittsburgh, PA) for whole-genome sequencing using the Illumina NextSeq 2000 platform (150 bp paired-end reads). Reads were mapped onto the *L. monocytogenes* 10403S reference sequence (RefSeq accession number NC_017544) using Bowtie2 (version 2.3.5.1) [82]. Single nucleotide polymorphisms (SNPs) were called using SAMtools and BCFtools (version 1.9) [83,84].

## Growth assays

*L. monocytogenes* strains were grown overnight (16–18 hours) in BHI at 30˚C on a slant. For BHI growth assays, cultures were inoculated 1:50 into 100 μl of BHI in 96-well plates. Optical density ($OD_{600}$) was measured every 15 minutes in an Eon Microplate Spectrophotometer (BioTek Instruments, Inc., Winooski, VT), and plates were incubated at 37˚C with 180 RPM shaking between time points. For *L. monocytogenes* MIC assays, cultures were inoculated 1:50 into 100 μl BHI containing serial 2-fold dilutions of the indicated compound in 96-well plates. Optical density ($OD_{600}$) was measured at 0 and 12 hours post-inoculation in a Synergy HT Microplate Spectrophotometer (BioTek Instruments, Inc., Winooski, VT), and plates were incubated on a 37˚C shaker platform (250 RPM) between time points. The MIC was defined as the lowest concentration of CRO at which the endpoint $OD_{600}$ minus the starting $OD_{600}$ was <0.1, the lowest $OD_{600}$ with visible turbidity. For *S. aureus* MIC assays, Clinical and Laboratory Standards Institute protocols were followed with minor modifications [85]. Frozen stocks of *S. aureus* were streaked onto fresh TSA and incubated at 37˚C for 15 hours. Single colonies were selected in biological duplicate, inoculated into 5 mL TSB, and grown for 15 h at 37˚C with 180 RPM shaking. Strains were back diluted 1:100 in 100 μl cation-adjusted Muller Hinton broth supplemented with 2% sodium chloride (CA-MH) containing 2-fold dilutions of oxacillin 96-well plates and incubated at 37˚C with 180 RPM shaking. After 12 hours, final $OD_{600}$ was measured in the Eon spectrophotometer.

## Bacteriolysis assays

Intracellular lysis of *L. monocytogenes* was measured by luciferase reporter activity as previously described [6]. Briefly, 5 x $10^5$ immortalized *Ifnar*[-/-] BMDMs were seeded into 24-well plates overnight. *L. monocytogenes* strains carrying the pBHE573 reporter construct [6] were grown overnight (16–18 hours) in BHI at 30˚C on a slant. Cells were infected at an MOI of 10. One hour post-infection, media was removed from cells and replaced with fresh media containing gentamicin. Six hours post-infection, cells were lysed with TNT buffer, and luciferase activity was measured using luciferin reagent as previously described [6]. Luminescence was measured in a Synergy HT Microplate Spectrophotometer (BioTek Instruments, Inc., Winooski, VT).

## LDH assays

Induction of host cell death by *L. monocytogenes* was measured by lactate dehydrogenase (LDH) release as previously described [86]. Briefly, 5 x $10^5$ BMDMs were seeded into 24-well plates overnight with 100 ng/ml Pam3CSK4 pre-treatment. *L. monocytogenes* strains were grown overnight (16–18 hours) in BHI at 30˚C on a slant. Cells were infected at an MOI of 10.

Half an hour post-infection, media was removed from cells and replaced with fresh media containing gentamicin and Pam3CSK4. Six hours post-infection, LDH release from infected cells was measured as previously described [87,88] and normalized to 100% lysis (determined by addition of Triton X-100 to a control well at a final concentration of 1%).

### Intracellular growth assays

Bone marrow-derived macrophages (BMDMs) were isolated from C57BL/6 mice as previously described [89]. BMDMs were seeded at 5 x 10$^6$ cells/60 mm dish with coverslips and allowed to adhere overnight. *L. monocytogenes* strains were grown overnight (16–18 hours) in BHI at 30˚C on a slant. BMDMs were infected at an MOI of 0.2. Half an hour post-infection, media was removed and replaced with fresh media containing gentamicin. The number CFU per coverslip was determined at the indicated time points post-infection as previously described [90]. Shown is a representative growth curve from three biological replicates.

### Plaquing assays

Plaque assays were conducted using the L2 cell line (murine fibroblasts [91]) essentially as previously described [92]. Briefly, 1.2 x 10$^6$ cells were seeded in 6-well plates overnight. *L. monocytogenes* strains were grown overnight (16–18 hours) in BHI at 30˚C on a slant. Cells were washed twice with pre-warmed PBS and then infected at an MOI of ~0.5 One hour post-infection, the media was removed, cells were washed thrice with pre-warmed PBS, and then fresh media containing gentamicin and 0.7% agarose was added to the wells. A second agarose plug was added ~48 hours post-infection. Three to five days post-infection, agarose plugs were removed, cells were stained with 0.3% crystal violet for 10 minutes, and wells were washed twice with double distilled water and allowed to dry completely. Stained wells were imaged, and plaque sizes were measured using ImageJ software (version 1.53).

### Peptidoglycan labeling *in vitro* and quantitation by flow cytometry

Peptidoglycan synthesis by *L. monocytogenes* grown in broth was measured by copper-catalyzed azide-alkyne cycloaddition (CuAAC) reactions essentially as previously described [63,64]. Cells growing exponentially in BHI were back-diluted to OD$_{600}$ 0.3 in 1 ml BHI medium, with or without the addition of 0.5 MIC of CRO (as determined in Table 4). 1 mM of alkyne-D-alanine-D-alanine (EDA-DA, [64]) was added and cultures were grown for 1 hour to mid-log phase (final OD$_{600}$ 0.5–0.9) at 37˚C with shaking. ~100 μl aliquots (normalized by OD$_{600}$) were washed twice with an equal volume PBS, fixed in ice-cold 70% ethanol for 10 minutes at -20˚C, and again washed twice with PBS. Cells were resuspended in 50 μl of CuAAC reaction mixture (128 μM TBTA, 1 mM CuSO$_4$, 1.2 mM freshly prepared sodium ascorbate, and 20 μM AFDye 488 Azide Plus fluorophore [Click Chemistry Tools] in PBS + 0.1% Triton X-100 + 0.01% BSA) and incubated with shaking at room temperature for 30 minutes in the dark. Cells were washed twice with PBS and finally resuspended in FACS buffer. Samples were acquired using an LSRII flow cytometer (BD Biosciences, San Jose, CA) with FACSDiva software (BD Biosciences) and analyzed using FlowJo software (Tree Star, Ashland, OR). Median fluorescence intensities (MFIs) are reported.

### Mouse infections

Intravenous mouse infections were performed as previously described [86]. *L. monocytogenes* strains were grown overnight (16–18 hours) in BHI at 30˚C on a slant. Overnight cultures were back-diluted 1:5 into BHI and grown at 37˚C with shaking to mid-log phase (OD$_{600}$ 0.4–

0.6). Cultures were then washed with PBS and adjusted to $5 \times 10^5$ CFU/ml. Nine-week-old C57BL/6 mice (Charles River Laboratories) were infected with $1 \times 10^5$ CFU in 200 µl PBS. Forty-eight hours post-infection, animals were euthanized by $CO_2$ asphyxiation according to AVMA standards. Spleens and livers were harvested, homogenized in PBS + 0.1% NP-40, and plated for CFU. Infections were performed in biological duplicate, one with female mice and one with male mice.

### *In vivo* suppressor screen

The $\Delta prkA$::*erm* mutant was mutagenized by a 3-minute exposure to ethyl methanesulfonate (EMS) following a protocol previously described [59,81]. One ml of the library was thawed, washed in PBS and resuspended in BHI, back-diluted 1:5 into BHI, and grown at 37˚C with shaking for one doubling (measured by $OD_{600}$). Cultures were then washed with PBS and adjusted to $5 \times 10^7$ CFU/ml. Twelve 24-week-old female C57BL/6 mice (Charles River Laboratories) were each infected intravenously with $1 \times 10^7$ CFU of this library in 200 µl PBS. Seventy-two hours post-infection, spleens were harvested, homogenized, and plated on BHI plates with streptomycin. Colonies from each mouse were picked into 96-well plates containing 100 µl BHI broth, grown overnight at 37˚C with shaking, pelleted by centrifugation, resuspended in 100 µl BHI + 40% glycerol, and stored at -80˚C. One isolate from the spleen of each mouse was subsequently streaked onto BHI plates with streptomycin, grown overnight in BHI broth, and subjected to whole genome sequencing and SNP analysis as described above.

### Statistics

Statistical analyses were performed with GraphPad Prism (version 9). Means from more than two groups were compared using one-way analysis of variance (ANOVA) or Kruskal-Wallis tests as appropriate with the post-tests indicated in the figure legends. For the phosphoproteomics experiments, *P* values were calculated using the Proteome Discoverer software (version 2.4.1.15). For bacteriolysis assays, the holin-lysin strain was not included in statistical analyses as it is an engineered control and is thus not biologically meaningful.

### Supporting information

**S1 Table. All phosphopeptides identified in wild type and the $\Delta prkA_{cond}$ mutant of *L. monocytogenes*, and PrkA-dependent phosphosites.**
(XLSX)

**S2 Table. All phosphopeptides identified in wild type and the $\Delta prkA$::*erm* mutant of *L. monocytogenes*.**
(XLSX)

**S3 Table. Serine and threonine phosphopeptides with XCorr scores $>$ 2.0 identified in wild type and the $\Delta prkA$::*erm* mutant of *L. monocytogenes*.**
(XLSX)

**S1 Fig. *prkA* is not an essential gene in *L. monocytogenes*.** (**A**) Growth of WT, the $\Delta prkA$::*erm* strain, and 10 $\Delta prkA$::*erm* transductants (T1-T10) in BHI was monitored by $OD_{600}$. Error bars indicate SD; n = 3. (**B**) Bars indicate median MICs of CRO for the indicated strains; n = 3. No statistical differences between strains were found between the transductants and $\Delta prkA$::*erm* by one-way ANOVA with Tukey's multiple comparisons test. (**C**) Growth of WT, the $\Delta prkA$::*erm* mutant, and $\Delta prkA$::*erm* carrying the pIMK2-*prkA* complementation construct in BHI was monitored by $OD_{600}$. Error bars indicate SD; n = 3. (**D**) Bars indicate median MICs

of CRO for the indicated strains; n = 3. *, $P < 0.05$ compared to wild type, and #, $P < 0.05$ for the indicated comparisons, by one-way ANOVA with Tukey's multiple comparisons test.
(TIF)

**S2 Fig. *Ex vivo* phenotypes of a ΔreoM mutant can be complemented by reintroduction of *reoM* controlled by its native promoter.** (**A**) Bars indicate median MICs of CRO for the indicated *L. monocytogenes* strains; n = 3. (**B**) Intracellular bacteriolysis in immortalized *Ifnar*^-/-^ macrophages. Macrophages were infected with the indicated strains carrying the pBHE573 reporter vector at an MOI of 10, and luciferase activity was measured 6 hours post-infection. Error bars indicate SEM; n = 5. Plaque formation in immortalized murine fibroblasts (L2 cells). L2s were infected with the indicated strains at an MOI of ~0.5, plaques were stained on day 4 of infection, and sizes were normalized to those of wild type. Error bars indicate SEM; data are averaged from a minimum of 64 plaques from three biological replicates. N.D., not detected. (**A**-**C**) *, $P < 0.05$ compared to wild type, and #, $P < 0.05$ for the indicated comparisons, by one-way ANOVA with Tukey's multiple comparisons test.
(TIF)

**S3 Fig. Bacterial burdens from *in vivo* suppressor screen.** CFU were enumerated 72 hours post-infection with the EMS-mutagenized library of the Δ*prkA*::*erm* mutant.
(TIF)

## Acknowledgments

We thank Grzegorz Sabat and Dr. Greg Barrett-Wilt of the UW-Madison Biotechnology Center for help planning, performing, and interpreting the mass spectrometry experiments. We acknowledge the Microbial Genome Sequencing (MiGS) Center for whole genome sequencing services. We thank Dr. M. Sloan Siegrist for help planning and providing reagents for the peptidoglycan labeling experiment. We thank Dr. Courtney McDougal for assistance performing flow cytometry and Emily Forster for constructing the pBHE573k vector.

## Author Contributions

**Conceptualization:** Jessica L. Kelliher, John-Demian Sauer.

**Formal analysis:** Jessica L. Kelliher, Caroline M. Grunenwald, Cassandra I. Lew, John-Demian Sauer.

**Funding acquisition:** Jessica L. Kelliher, Caroline M. Grunenwald, Warren E. Rose, John-Demian Sauer.

**Investigation:** Jessica L. Kelliher, Caroline M. Grunenwald, Rhiannon R. Abrahams, McKenzie E. Daanen.

**Writing – original draft:** Jessica L. Kelliher, John-Demian Sauer.

**Writing – review & editing:** Jessica L. Kelliher, Caroline M. Grunenwald, Warren E. Rose, John-Demian Sauer.

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
