## [Decision Letter · Decision Letter 0]

31 Aug 2021

Dear Dr. Sauer,

Thank you very much for submitting your manuscript "PASTA kinase-dependent control of peptidoglycan synthesis via ReoM is required for cell wall stress responses, cytosolic survival, and virulence in Listeria monocytogenes" for consideration at PLOS Pathogens. As with all papers reviewed by the journal, your manuscript was reviewed by members of the editorial board and by several independent reviewers. The reviewers appreciated the attention to an important topic. Based on the reviews, we are likely to accept this manuscript for publication, providing that you modify the manuscript according to the review recommendations.  Specifically, we ask that you consider altering the text to account for the suggestions and concerns of the reviewers.  

Sincerely,

Christopher M. Sassetti

Associate Editor

PLOS Pathogens

Raphael Valdivia

Section Editor

PLOS Pathogens

Kasturi Haldar

Editor-in-Chief

PLOS Pathogens

orcid.org/0000-0001-5065-158X

Michael Malim

Editor-in-Chief

PLOS Pathogens

orcid.org/0000-0002-7699-2064

Reviewer Comments (if any, and for reference):

Reviewer's Responses to Questions

**Part I - Summary**

Reviewer #1: Using complementary methods, Kelliher et al. identify and validate ReoM as an important PrkA phosphosubstrate in vitro during peptidoglycan stress and likely in vivo. Their work raises fascinating questions about pathogen cell wall biogenesis and repair during different stages of infection, both at the cell and whole-animal levels. The experiments are rigorous and their claims are well-supported.

Reviewer #2: The manuscript by Kelliher and co-workers describes studies of the PrkA-ReoM signaling pathway in Listeria. They extend previous findings from both their group and that of Wamp et al to document that the PrkA-ReoM signaling pathway seems to be essential for infection and virulence by L. monocytogenes, and plays a role in antimicrobial resistance in S. aureus as well. They also report that the PrkA gene is not essential in L. monocytogenes, a finding that contradicts previous published reports. With a few exceptions described in detail below, the work is well done overall and the manuscript is clearly written.

Reviewer #3: The manuscript by Kelliher and co-workers titled “PASTA kinase-dependent control of peptidoglycan synthesis via ReoM is required for cell wall stress responses, cytosolic survival, and virulence in Listeria monocytogenes” is dealing with an important topic that involves regulation of peptidoglycan synthesis and its impact on virulence, in this case of the bacterial pathogen Listeria monocytogenes. The research was conducted by a group that carried out several studies on the impact of the serine/threonine associated kinase PrkA on L. monocytogenes virulence and cytosolic survival.

Unfortunately for this group, a paper was published in 2020 by Wamp et al., that similarly identifies ReoM as PrkA substrate. This paper also deciphered the function of these proteins in PG synthesis, demonstrating that PrkA and ReoM control the stability of MurA, which is an important enzyme in PG synthesis.

That said, the study performed here is beautiful, combining phosphoproteomic and genetic screens to search for PrkA substrates (under cell wall stress). Moreover, this study, in contrast to the one by Wamp, shows the effect of PrkA and ReoM on L. monocytogenes virulence. The paper is very well written and the results support the main conclusions. It provides new information regarding the regulation of PG synthesis during Lm infection of mammalian cells, and hence to my opinion justifies a publication in PLoS Pathogens.

**Part II – Major Issues: Key Experiments Required for Acceptance**

Reviewer #1: No major issues

Reviewer #2: Figure 3, and lines 323, 342, 350-1, 352-3, 354-5, 491-3 and elsewhere: The authors use an innovative click-chemistry technology for labeling of PG of different strains in the presence/absence of antimicrobial treatment. Although this labeling technique is powerful, my main concern is that the authors equate the extent of such labeling to the amount of PG synthesis, and this interpretation does not seem justified given the available data. The labeling strategy can provide insights into “where” in the cell the PG is, or “when” it is synthesized (depending on the experimental setup), but extending it to “how much” PG is synthesized is a bit tricky. This is largely due to the fact that PG is dynamic, and the amount of PG at any given time (i.e. the amount of labeling observed) depends on both how much is made as well as the turnover (i.e. loss) as well as the efficiency of label incorporation. For example, if the label is incorporated more efficiently into nascent PG under a given set of conditions (e.g. a mutant strain, or drug treatment), then the labeling signal will increase even if the amount of PG has not. There are numerous reasons that the efficiency of label incorporation could increase (e.g. enhanced turnover of existing PG through hydrolytic enzymes, enhanced uptake of metabolic label into cells, etc), and these have not been experimentally addressed in the manuscript. In short, the authors stated conclusions that prkA / reoM control the amount of PG synthesis is one plausible possibility, but it is not rigorously supported by the available data.

Figure 3: Related to the above point, the growth kinetics / growth stage of the strains for the PG labeling study is not clear from the description given, and could impact the outcome. Do all the strains analyzed grow at the same rate? Were they in the same stage of growth at the time of sampling? Although it is a bit hard to tell, based on the growth curve shown in Supp Fig 1 and the description in the methods (lines 765-7), one might imagine the strains (untreated wild-type, at least) would nearly be entering stationary phase at the time of sampling/harvest, which complicates things because it might be expected that PG synthesis would be slowing during that time. If the drug-treated cells or mutant strain is growing more slowly and not yet entering stationary phase at the time of sampling, PG synthesis could be inherently higher simply for that reason.

line 393: This statement is a bit strongly worded given the data presented. Strictly speaking, the authors analyzed strains lacking reoM or prkA entirely, and did not specifically demonstrate that ReoM is phosphorylated (by PrkA or otherwise) during infection. While it is clear that loss of reoM has phenotypic consequences during infection, based on this data it does not seem justified to specifically attribute those consequences to PrkA-dependent phosphorylation of ReoM during infection. A similar argument applies to the S. aureus data and conclusion (line 433).

Reviewer #3: I don't have such experiments.

**Part III – Minor Issues: Editorial and Data Presentation Modifications**

Reviewer #1: • Author Summary has technical language that should be simplified and/or explained more, e.g., phosphoproteomics

• P. 11 lines 237-238: I believe the samples were harvested only from drug-treated wild-type and prkA mutant. In the absence of untreated controls, it is not quite accurate to say that PrkA mediates a global response to cell wall stress since we do not know what wt vs. mutant looks like in the absence of that stress.

• P. 12 line 266: I know there is a reference, but it would be helpful to the reader to include a very brief summary of how the reporter system works.

• P. 13/14 lines 288-290: I think it’s a bit more nuanced than the concluding sentence implies. Suppressor phenotypes in panels A and B correlate well but less true for panels D and E. This is explained well in the preceding paragraph, although I disagree that the parent prkA mutant is “cleared” in Fig. 2D as described in line 276.

• Fig. 3B labels hard to read.

• P. 26 lines 569-570: likely to be true but phosphoproteomics performed in vitro

• A model cartoon would augment the discussion. I see that Wamp et al. have one, but it would be helpful to visualize the signaling pathway within a broader, host-pathogen context, especially since this is a key aspect of the current work.

Reviewer #2: lines 197-211: The conclusion that prkA is not essential is based on reasonably compelling analyses (whole genome sequencing and phage transduction studies). But, whole genome sequencing can miss genomic variants due to gaps or low coverage in some regions, and phage transduction could co-transduce a secondary mutation that is physically close to the prkA::erm allele in the genome. Given the contradiction of the current results with those published in the literature previously (i.e. that prkA is essential), why not just do a complementation assay by expressing prkA in trans in the prkA::erm mutant to demonstrate a rescue of the growth defect and put the final nail in the coffin on this? It appears the authors possess a plasmid with inducible prkA that would allow them to do this.

Figure 6: Would have been nice to see some complementation data for the S. aureus mutants described here.

Tables 4 and 5: It is unclear how nucleotide variants identified in intergenic regions can be assigned to particular genes without experimental analysis (e.g. PCS1 murZ [Table 4], PEVS3 guaA [Table 5], others). For example, it isn’t hard to imagine that such intergenic variants could affect some regulatory element (e.g. small RNA) that is not related to the adjacent gene.

Reviewer #3: Minor comments:

In order to better deduce the role of PrkA in phosphorylation of target proteins, the authors performed a comprehensive analysis of multiple phosphosites of L. monocytogenes proteins. I expect this data to be interesting to researchers dealing with Listeria, and hence I would like to recommend the authors to slightly edit the presentation of both Supplementary Tables (1 and 2). Currently the data within these tables are associated with the use of Reference Proteins whose accession numbers have a universal prefix WP_. From my experience, it is not that intuitive to ‘translate’ any WP_ accession number to a real gene tag of the 10403S genome as well as to that of strain EGDe. It would be useful if the authors can add an additional column that provides real gene tags of strains 10403S and EGDe.

I wonder if the authors can comment on the possible roles of the other two proteins encoded by the operon of reoM. lmo1502 (LMRG_01468) and lmo1501 (LMRG_01469) located immediately downstream to reoM (lmo1503, LMRG_01467). Have the authors analyzed any feedback regulatory changes of these genes in the delta reoM mutant? According to Wurtzel et al (2012) this operon has a high expression level and it would be interesting to learn how it is regulated? lmo1502 encodes a protein resembling RNase H, while lmo1501 encodes a conserved acidic protein (pI 3.59) of unknown function (DUF1292; pfam06949). Are they important for ReoM expression/activity?

I would suggest another presentation of the PSC results; first sequencing the mutants, then basic characterization of all mutants, and then focus on three mutants of your choice.

Figure 3B- unreadable and should be presented as a table.

PLOS authors have the option to publish the peer review history of their article (what does this mean?). If published, this will include your full peer review and any attached files.

Reviewer #1: No

Reviewer #2: No

Reviewer #3: **Yes: **Anat A. Herskovits

Figure Files:

Data Requirements:

Reproducibility:

References:

---

## [Editor Report · Decision Letter 1]

27 Sep 2021

Dear Dr. Sauer,

We are pleased to inform you that your manuscript 'PASTA kinase-dependent control of peptidoglycan synthesis via ReoM is required for cell wall stress responses, cytosolic survival, and virulence in Listeria monocytogenes' has been provisionally accepted for publication in PLOS Pathogens.

Best regards,

Christopher M. Sassetti

Associate Editor

PLOS Pathogens

Raphael Valdivia

Section Editor

PLOS Pathogens

Kasturi Haldar

Editor-in-Chief

PLOS Pathogens

orcid.org/0000-0001-5065-158X

Michael Malim

Editor-in-Chief

PLOS Pathogens

orcid.org/0000-0002-7699-2064
---

## [Editor Report · Acceptance letter]

4 Oct 2021

Dear Dr. Sauer,

We are delighted to inform you that your manuscript, "PASTA kinase-dependent control of peptidoglycan synthesis via ReoM is required for cell wall stress responses, cytosolic survival, and virulence in Listeria monocytogenes," has been formally accepted for publication in PLOS Pathogens.

Best regards,

Kasturi Haldar

Editor-in-Chief

PLOS Pathogens

orcid.org/0000-0001-5065-158X

Michael Malim

Editor-in-Chief

PLOS Pathogens

orcid.org/0000-0002-7699-2064